# RECON: REDUCING CONFLICTING GRADIENTS FROM THE ROOT FOR MULTI-TASK LEARNING

**Guangyuan Shi, Qimai Li, Wenlong Zhang, Jiaxin Chen, Xiao-Ming Wu**[✉]
Department of Computing, The Hong Kong Polytechnic University, Hong Kong S.A.R., China
{guang-yuan.shi, qee-mai.li, wenlong.zhang}@connect.polyu.hk,
jiax.chen@connect.polyu.hk, xiao-ming.wu@polyu.edu.hk

## ABSTRACT

A fundamental challenge for multi-task learning is that different tasks may conflict with each other when they are solved jointly, and a cause of this phenomenon is *conflicting gradients* during optimization. Recent works attempt to mitigate the influence of conflicting gradients by directly altering the gradients based on some criteria. However, our empirical study shows that "gradient surgery" cannot effectively reduce the occurrence of conflicting gradients. In this paper, we take a different approach to reduce conflicting gradients *from the root*. In essence, we investigate the task gradients w.r.t. each *shared* network layer, select the layers with high conflict scores, and turn them to *task-specific* layers. Our experiments show that such a simple approach can greatly reduce the occurrence of conflicting gradients in the remaining shared layers and achieve better performance, with only a slight increase in model parameters in many cases. Our approach can be easily applied to improve various state-of-the-art methods including gradient manipulation methods and branched architecture search methods. Given a network architecture (e.g., ResNet18), it only needs to search for the conflict layers once, and the network can be modified to be used with different methods on the same or even different datasets to gain performance improvement. The source code is available at https://github.com/moukamisama/Recon.

## 1 INTRODUCTION

**Multi-task learning** (MTL) is a learning paradigm in which multiple different but correlated tasks are jointly trained with a shared model (Caruana, 1997), in the hope of achieving better performance with an overall smaller model size than learning each task independently. By discovering shared structures across tasks and leveraging domain-specific training signals of related tasks, MTL can achieve efficiency and effectiveness. Indeed, MTL has been successfully applied in many domains including natural language processing (Hashimoto et al., 2017), reinforcement learning (Parisotto et al., 2016; D'Eramo et al., 2020) and computer vision (Vandenhende et al., 2021).

**A major challenge** for multi-task learning is *negative transfer* (Ruder, 2017), which refers to the performance drop on a task caused by the learning of other tasks, resulting in worse overall performance than learning them separately. This is caused by *task conflicts*, i.e., tasks compete with each other and unrelated information of individual tasks may impede the learning of common structures. From the optimization point of view, a cause of negative transfer is *conflicting gradients* (Yu et al., 2020), which refers to two task gradients pointing away from each other and the update of one task will have a negative effect on the other. Conflicting gradients make it difficult to optimize the multi-task objective, since task gradients with larger magnitude may dominate the update vector, making the optimizer prioritize some tasks over others and struggle to converge to a desirable solution.

**Prior works** address task/gradient conflicts mainly by balancing the tasks via task reweighting or gradient manipulation. Task reweighting methods adaptively re-weight the loss functions by homoscedastic uncertainty (Kendall et al., 2018), balancing the pace at which tasks are learned Chen et al. (2018); Liu et al. (2019), or learning a loss weight parameter (Liu et al., 2021b). Gradient manipulation methods reduce the influence of conflicting gradients by directly altering the gradients based on different criteria (Sener & Koltun, 2018; Yu et al., 2020; Chen et al., 2020; Liu et al.,

2021a) or rotating the shared features (Javaloy & Valera, 2022). While these methods have demonstrated effectiveness in different scenarios, in our empirical study, we find that they cannot reduce the occurrence of conflicting gradients (see Sec. 3.3 for more discussion).

**We propose a different approach** to reduce conflicting gradients for MTL. Specifically, we investigate layer-wise conflicting gradients, i.e., the task gradients w.r.t. each shared network layer. We first train the network with a regular MTL algorithm (e.g., joint-training) for a number of iterations, compute the conflict scores for all shared layers, and select those with highest conflict scores (indicating severe conflicts). We then set the selected shared layers task-specific and train the modified network from scratch. As demonstrated by comprehensive experiments and analysis, our simple approach Recon has the following key advantages: **(1)** Recon can greatly reduce conflicting gradients with only a slight increase in model parameters (less than 1% in some cases) and lead to significantly better performance. **(2)** Recon can be easily applied to improve various gradient manipulation methods and branched architecture search methods. Given a network architecture, it only needs to search for the conflict layers once, and the network can be modified to be used with different methods and even on different datasets to gain performance improvement. **(3)** Recon can achieve better performance than branched architecture search methods with a much smaller model.

## 2 RELATED WORKS

In this section, we briefly review related works in multi-task learning in four categories: tasks clustering, architecture design, architecture search, and task balancing. *Tasks clustering methods* mainly focus on identifying which tasks should be learned together (Thrun & O'Sullivan, 1996; Zamir et al., 2018; Standley et al., 2020; Shen et al., 2021; Fifty et al., 2021).

*Architecture design methods* include hard parameter sharing methods (Kokkinos, 2017; Long et al., 2017; Bragman et al., 2019), which learn a shared feature extractor and task-specific decoders, and soft parameters sharing methods (Misra et al., 2016; Ruder et al., 2019; Gao et al., 2019; 2020; Liu et al., 2019), where some parameters of each task are assigned to do cross-task talk via a sharing mechanism. Compared with soft parameters sharing methods, our approach Recon has much better scalability when dealing with a large number of tasks.

Instead of designing a fixed network structure, some methods (Rosenbaum et al., 2018; Meyerson & Miikkulainen, 2018; Yang et al., 2020) propose to dynamically self-organize the network for different tasks. Among them, *branched architecture search* (Guo et al., 2020; Bruggemann et al., 2020) methods are more related to our work. They propose an automated architecture search algorithm to build a tree-structured network by learning where to branch. In contrast, our method Recon decides which layers to be shared across tasks by considering the severity of layer-wise conflicting gradients, resulting in a more compact architecture with lower time cost and better performance.

Another line of research is *task balancing* methods. To address task/gradient conflicts, some methods attempt to re-weight the multi-task loss function using homoscedastic uncertainty (Kendall et al., 2018), task prioritization (Guo et al., 2018), or similar learning pace (Liu et al., 2019; 2021b). GradNorm (Chen et al., 2018) learns task weights by dynamically tuning gradient magnitudes. MGDA (Sener & Koltun, 2018) find the weights by minimizing the norm of the weighted sum of task gradients. To reduce the influence of conflicting gradients, PCGrad (Yu et al., 2020) projects each gradient onto the normal plane of another gradient and uses the average of projected gradients for update. Graddrop (Chen et al., 2020) randomly drops some elements of gradients based on element-wise conflict. CAGrad (Liu et al., 2021a) ensures convergence to a minimum of the average loss across tasks by gradient manipulation. RotoGrad (Javaloy & Valera, 2022) re-weights task gradients and rotates the shared feature space. Instead of manipulating gradients, our method Recon leverages gradient information to modify network structure to mitigate task conflicts from the root.

## 3 PILOT STUDY: TASK CONFLICTS IN MULTI-TASK LEARNING

### 3.1 MULTI-TASK LEARNING: PROBLEM DEFINITION

Multi-task learning (MTL) aims to learn a set of correlated tasks $\{\mathcal{T}_i\}_{i=1}^T$ simultaneously. For each task $\mathcal{T}_i$, the empirical loss function is $\mathcal{L}_i(\theta_{\mathrm{sh}}, \theta_i)$, where $\theta_{\mathrm{sh}}$ are parameters shared among all tasks

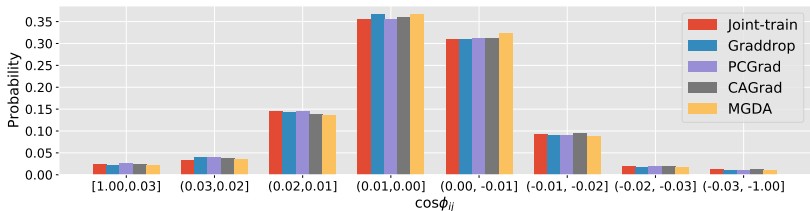

Figure 1: The distributions of gradient conflicts (in terms of $\cos \phi_{ij}$) of the joint-training baseline and state-of-the-art gradient manipulation methods on Multi-Fashion+MNIST benchmark.

and $\theta_i$ are task-specific parameters. The goal is to find optimal parameters $\theta = \{\theta_{\text{sh}}, \theta_1, \theta_2, \cdots, \theta_T\}$ to achieve high performance across all tasks. Formally, it aims to minimize a multi-task objective:

$$\theta^* = \arg \min_\theta \sum_i^T w_i \mathcal{L}_i(\theta_{\text{sh}}, \theta_i), \tag{1}$$

where $w_i$ are pre-defined or dynamically computed weights for different tasks. A popular choice is to use the average loss (i.e., equal weights). However, optimizing the multi-task objective is difficult, and a known cause is conflicting gradients.

### 3.2 CONFLICTING GRADIENTS

Let $\mathbf{g}_i = \nabla_{\theta_{\text{sh}}} \mathcal{L}_i(\theta_{\text{sh}}, \theta_i)$ denote the gradient of task $\mathcal{T}_i$ w.r.t. the shared parameters $\theta_{\text{sh}}$ (i.e., a vector of the partial derivatives of $\mathcal{L}_i$ w.r.t. $\theta_{\text{sh}}$) and $g_i^{\text{ts}} = \nabla_{\theta_i} \mathcal{L}_i(\theta_{\text{sh}}, \theta_i)$ denote the gradient w.r.t. the task-specific parameters $\theta_i$. A small change of $\theta_{\text{sh}}$ in the direction of negative $\mathbf{g}_i$ is $\theta_{\text{sh}} \leftarrow \theta_{\text{sh}} - \alpha \mathbf{g}_i$, with a sufficiently small step size $\alpha$. The effect of this change on the performance of another task $\mathcal{T}_j$ is measured by:

$$\Delta \mathcal{L}_j = \mathcal{L}_j(\theta_{\text{sh}} - \alpha \mathbf{g}_i, \theta_j) - \mathcal{L}_j(\theta_{\text{sh}}, \theta_j) = -\alpha \mathbf{g}_i \cdot \mathbf{g}_j + o(\alpha), \tag{2}$$

where the second equality is obtained by first order Taylor approximation. Likewise, the effect of a small update of $\theta_{\text{sh}}$ in the direction of the negative gradient of task $\mathcal{T}_j$ (i.e., $-\mathbf{g}_j$) on the performance of task $\mathcal{T}_i$ is $\Delta \mathcal{L}_i = -\alpha \mathbf{g}_i \cdot \mathbf{g}_j + o(\alpha)$. Notably, the model update for task $\mathcal{T}_i$ is considered to have a negative effect on task $\mathcal{T}_j$ when $\mathbf{g}_i \cdot \mathbf{g}_j < 0$, since it increases the loss of task $\mathcal{T}_j$, and vice versa. A formal definition of conflicting gradients is given as follows (Yu et al., 2020).

**Definition 1** (Conflicting Gradients). *The gradients $\mathbf{g}_i$ and $\mathbf{g}_j (i \neq j)$ are said to be conflicting with each other if $\cos \phi_{ij} < 0$, where $\phi_{ij}$ is the angle between $\mathbf{g}_i$ and $\mathbf{g}_j$.*

As shown in Yu et al. (2020), conflicts in gradient pose serious challenges for optimizing the multi-task objective (Eq. 1). Using the average gradient (i.e., $\frac{1}{T}\sum_{i=1}^T \mathbf{g}_i$) for gradient decent may hurt the performance of individual tasks, especially when there is a large difference in gradient magnitudes, which will make the optimizer struggle to converge to a desirable solution.

### 3.3 GRADIENT SURGERY CANNOT EFFECTIVELY REDUCE CONFLICTING GRADIENTS

To mitigate the influence of conflicting gradients, several methods (Yu et al., 2020; Chen et al., 2020; Liu et al., 2021a) have been proposed to perform "gradient surgery". Instead of following the average gradient direction, they alter conflicting gradients based on some criteria and use the modified gradients for model update. We conduct a pilot study to investigate whether gradient manipulation can effectively reduce the occurrence of conflicting gradients. For each training iteration, we first calculate the task gradients of all tasks w.r.t. the shared parameters (i.e., $\mathbf{g}_i$ for any task $i$) and compute the conflict angle between any two task gradients $\mathbf{g}_i$ and $\mathbf{g}_j$ in terms of $cos\phi_{ij}$. We then count and draw the distribution of $cos\phi_{ij}$ in all training iterations. We provide the statistics of the joint-training baseline (i.e., training all tasks jointly with equal loss weights and all parameters shared) and several state-of-the-art gradient manipulation methods including GradDrop (Chen et al., 2020), PCGrad (Yu et al., 2020), CAGrad (Liu et al., 2021a), and MGDA (Sener & Koltun, 2018) on Multi-Fashion+MNIST (Lin et al., 2019), CityScapes, NYUv2, and PASCAL-Context datasets.

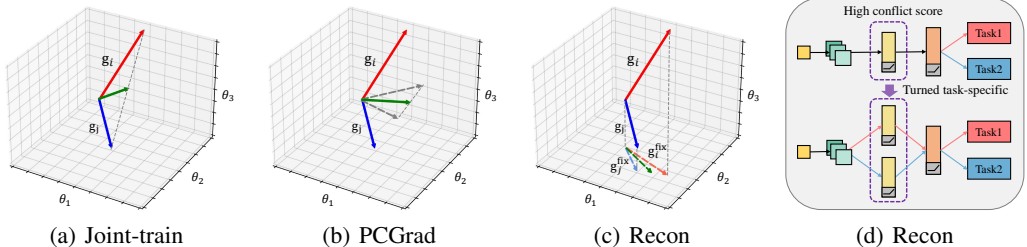

(a) Joint-train     (b) PCGrad     (c) Recon     (d) Recon

Figure 2: Illustration of the differences between joint-training, gradient manipulation, and our approach. (a) In joint-training, the update vector (in green) is the average gradient $\frac{1}{2}(\mathbf{g}_i + \mathbf{g}_j)$. Due to the conflict between $\mathbf{g}_i$ and $\mathbf{g}_j$, the update vector is dominated by $\mathbf{g}_i$ (in red). (b) PCGrad (Yu et al., 2020) projects each gradient onto the normal plane of the other one and uses the average of the projected gradients (indicated by dashed grey arrows) as the update vector (in green). As such, the update vector is less dominated by $\mathbf{g}_i$. (c) Our approach Recon finds the parameters contributing most (e.g., $\theta_3$) to gradient conflicts and turns them into task specific ones. In effect, it performs an orthographic/coordinate projection of conflicting gradients to the space of the rest parameters (e.g., $\theta_1$ and $\theta_2$) such that the projected gradients $\mathbf{g}_i^{\text{fix}}$ and $\mathbf{g}_j^{\text{fix}}$ are better aligned. (d) Illustration of Recon turning a shared layer with high conflict score to task-specific layers.

The results are provided in Fig. 1, Fig. 5, Fig. 6, Fig. 7, Table 6, and Tables 8-10. It can be seen that gradient manipulation methods can only slightly reduce the occurrence of conflicting gradients (compared to joint-training) in some cases, and in some other cases they even increase it.

## 4   OUR APPROACH: REDUCING CONFLICTING GRADIENTS FROM THE ROOT

Our pilot study shows that adjusting gradients for model update cannot effectively prevent the occurrence of conflicting gradients in MTL, which suggests that the root causes of this phenomenon may be closely related to the nature of different tasks and the way how model parameters are shared among them. Therefore, to mitigate task conflicts for MTL, in this paper, we take a different approach to reduce the occurrence of conflicting gradients from the root.

### 4.1   RECON: REMOVING LAYER-WISE CONFLICTING GRADIENTS

Our approach is extremely simple and intuitive. We first identify the shared network layers where conflicts occur most frequently and then turn them into task-specific parameters. Suppose the shared model parameters $\theta_{\text{sh}}$ are composed of $n$ layers, i.e., $\theta_{\text{sh}} = \{\theta_{\text{sh}}^{(k)}\}_{k=1}^n$, where $\theta_{\text{sh}}^{(k)}$ is the $k^{\text{th}}$ shared layer. Let $\mathbf{g}_i^{(k)}$ denote the gradient of task $\mathcal{T}_i$ w.r.t. the $k^{\text{th}}$ shared layer $\theta_{\text{sh}}^{(k)}$, i.e., $\mathbf{g}_i^{(k)}$ is a vector of the partial derivatives of $\mathcal{L}_i$ w.r.t. the parameters of $\theta_{\text{sh}}^{(k)}$. Let $\phi_{ij}^{(k)}$ denote the angle between $\mathbf{g}_i^{(k)}$ and $\mathbf{g}_j^{(k)}$. We define layer-wise conflicting gradients and $S$-conflict score as follows.

**Definition 2** (Layer-wise Conflicting Gradients). *The gradients $\mathbf{g}_i^{(k)}$ and $\mathbf{g}_j^{(k)}$ ($i \neq j$) are said to be conflicting with each other if $\cos \phi_{ij}^{(k)} < 0$.*

**Definition 3** ($S$-Conflict Score). *For any $-1 < S \leq 0$, the $S$-conflict score for the $k^{\text{th}}$ shared layer is the number of different pairs $(i, j)(i \neq j)$ s.t. $\cos \phi_{ij}^{(k)} < S$, denoted as $s^{(k)}$.*

$S$ indicates the severity of conflicts, and setting $S$ smaller means we care about cases of more severe conflicts. The $S$-conflict score $s^{(k)}$ indicates the occurrence of conflicting gradients at severity level $S$ for the $k^{\text{th}}$ shared layer. If $s^{(k)} = \binom{T}{2}$, it means that for any two different tasks, there is a conflict in their gradients w.r.t. the $k^{\text{th}}$ shared layer. By computing $S$-conflict scores, we can identify the shared layers where conflicts occur most frequently.

We describe our method Recon in Algorithm 1. First, we train the network for $I$ iterations and compute $S$-conflict scores for each shared layer $\theta^{(k)}$ in every iteration, denoted by $\{s_i^{(k)}\}_{i=1}^I$. Then,

---

**Algorithm 1:** Recon: Removing Layer-wise Conflicting Gradients

---

**Input:** Model parameters $\theta$, learning rate $\alpha$, a set of tasks $\{\mathcal{T}_i\}_{i=1}^{T}$, number of iterations $I$ for
computing conflict scores, conflict severity level $S$, number of selected layers $K$.

// Train the network and compute conflict scores for all layers

**for** iteration $i = 1, 2, \ldots, I$ **do**

    **for** $i = 1, 2, \ldots, T$ **do**

        Compute the gradients of task $\mathcal{T}_i$ w.r.t. all shared layers, i.e., $\{\mathbf{g}_i^{(k)}\}_{k=1}^{n}$ ;

    **end**

    Calculate the $S$-conflict scores for all shared layers in the current iteration, i.e., $\{s_i^{(k)}\}_{k=1}^{n}$;

    Update $\theta$ with joint-training or any gradient manipulation method ;

**end**

// Set layers with top conflict scores task-specific

For each layer $k$, calculate the sum of $S$-conflict scores in all iterations, i.e., $s^{(k)} = \sum_{i=1}^{I} s_i^{(k)}$;

Select the top $K$ layers with highest $s^{(k)}$ and set them task-specific;

// Train the modified network from scratch

**for** iteration $i = 1, 2, \ldots$ **do**

    Update $\theta$ with joint-training or any gradient manipulation method;

**end**

**Output:** Model parameters $\theta$.

---

we sum up the scores in all iterations, i.e., $s^{(k)} = \sum_{i=1}^{I} s_i^{(k)}$, and find the layers with highest $s^{(k)}$ scores. Next, we set these layers to be task-specific and train the modified network from scratch. We demonstrate the effectiveness of Recon by a theoretical analysis in Sec. 4.2 and comprehensive experiments in Sec. 5. The results show that Recon can effectively reduce the occurrence of conflicting gradients in the remaining shared layers and lead to substantial improvements over state-of-the-art.

## 4.2 THEORETICAL ANALYSIS

Here, we provide a theoretical analysis of Recon. Let $\theta_{\mathrm{sh}} = \{\theta_{\mathrm{sh}}^{\mathrm{fix}}, \theta_{\mathrm{sh}}^{\mathrm{cf}}\}$, where $\theta_{\mathrm{sh}}^{\mathrm{fix}}$ are the remaining shared parameters, and $\theta_{\mathrm{sh}}^{\mathrm{cf}}$ are those that will be turned to task-specific parameters $\theta_1^{\mathrm{cf}}, \theta_2^{\mathrm{cf}}, \cdots, \theta_T^{\mathrm{cf}}$. Notice that $\theta_1^{\mathrm{cf}}, \theta_2^{\mathrm{cf}}, \cdots, \theta_T^{\mathrm{cf}}$ will all be initialized with $\theta_{\mathrm{sh}}^{\mathrm{cf}}$. Therefore, after applying Recon, the model parameters are $\theta_r = \{\theta_{\mathrm{sh}}^{\mathrm{fix}}, \theta_1^{\mathrm{cf}}, \ldots, \theta_T^{\mathrm{cf}}, \theta_1^{\mathrm{ts}}, \ldots, \theta_T^{\mathrm{ts}}\}$. An one-step gradient update of $\theta_r$ is:

$$\hat{\theta}_{\mathrm{sh}}^{\mathrm{fix}} = \theta_{\mathrm{sh}}^{\mathrm{fix}} - \alpha \sum_{i=1}^{T} w_i \mathbf{g}_i^{\mathrm{fix}}, \quad \hat{\theta}_i^{\mathrm{cf}} = \theta_i^{\mathrm{cf}} - \alpha \mathbf{g}_i^{\mathrm{cf}}, \quad \hat{\theta}_i^{\mathrm{ts}} = \theta_i^{\mathrm{ts}} - \alpha \mathbf{g}_i^{\mathrm{ts}}, \quad i = 1, \ldots, T, \quad (3)$$

where $w_i$ are weight parameters, $\mathbf{g}_i^{\mathrm{ts}} = \nabla_{\theta_i^{\mathrm{ts}}} \mathcal{L}_i$, $\mathbf{g}_i^{\mathrm{cf}} = \nabla_{\theta_{\mathrm{sh}}^{\mathrm{cf}}} \mathcal{L}_i$ and $\mathbf{g}_i^{\mathrm{fix}} = \nabla_{\theta_{\mathrm{sh}}^{\mathrm{fix}}} \mathcal{L}_i$. Notice that different methods such as joint-training, MGDA Sener & Koltun (2018), PCGrad Yu et al. (2020), and CAGrad Liu et al. (2021a) choose different $w_i$ dynamically.

Without applying Recon, the model parameters are $\theta = \{\theta_{\mathrm{sh}}^{\mathrm{fix}}, \theta_{\mathrm{sh}}^{\mathrm{cf}}, \theta_1^{\mathrm{ts}}, \ldots, \theta_T^{\mathrm{ts}}\}$. An one-step gradient update of $\theta$ is given by

$$\hat{\theta}_{\mathrm{sh}}^{\mathrm{fix}} = \theta_{\mathrm{sh}}^{\mathrm{fix}} - \alpha \sum_{i=1}^{T} w_i \mathbf{g}_i^{\mathrm{fix}}, \quad \hat{\theta}_{\mathrm{sh}}^{\mathrm{cf}} = \theta_{\mathrm{sh}}^{\mathrm{cf}} - \alpha \sum_{i=1}^{T} w_i \mathbf{g}_i^{\mathrm{cf}}, \quad \hat{\theta}_i^{\mathrm{ts}} = \theta_i^{\mathrm{ts}} - \alpha \mathbf{g}_i^{\mathrm{ts}}, \quad i = 1, \ldots, T. \quad (4)$$

After the one-step updates, the loss functions with the updated parameters $\hat{\theta}_r$ and $\hat{\theta}$ respectively are:

$$\mathcal{L}(\hat{\theta}_r) = \sum_{i=1}^{T} \mathcal{L}_i \left( \hat{\theta}_{\mathrm{sh}}^{\mathrm{fix}}, \hat{\theta}_i^{\mathrm{cf}}, \hat{\theta}_i^{\mathrm{ts}} \right), \text{ and, } \mathcal{L}(\hat{\theta}) = \sum_{i=1}^{T} \mathcal{L}_i \left( \hat{\theta}_{\mathrm{sh}}^{\mathrm{fix}}, \hat{\theta}_{\mathrm{sh}}^{\mathrm{cf}}, \hat{\theta}_i^{\mathrm{ts}} \right), \quad (5)$$

where $\mathcal{L}_i$ is the loss function of task $\mathcal{T}_i$. Denote the set of indices of the layers turned task-specific by $\mathbb{P}$, then $\theta_{\mathrm{sh}}^{\mathrm{cf}} = \{\theta_{\mathrm{sh}}^{(k)}\}, k \in \mathbb{P}$. Assume that $\sum_{i=1}^{T} w_i = 1$, then we have the following theorem.

Table 1: Multi-task learning results on Multi-Fashion+MNIST dataset. All experiments are repeated over **3** random seeds and the mean values are reported. $\Delta m\%$ denotes the average relative improvement of all tasks. #P denotes model size (MB). The grey cell color indicates that Recon improves the result of the base model. The best average result is marked in bold.

| Method | Single-task | RotoGrad | BMTAS | Joint-train | w/ Recon | MGDA | w/ Recon | PCGrad | w/ Recon | GradDrop | w/ Recon | CAGrad | w/ Recon | MMoE | w/ Recon |
|---|---|---|---|---|---|---|---|---|---|---|---|---|---|---|---|
| T1 Acc↑ | 98.37 | 98.10 | 98.20 | 97.42 | 98.13 | 95.19 | **98.33** | 97.37 | 98.30 | 97.38 | 98.25 | 97.47 | 98.28 | 98.27 | 98.25 |
| T2 Acc↑ | 89.63 | 88.25 | 89.71 | 88.82 | 89.26 | 89.46 | 89.28 | 88.68 | **89.77** | 88.57 | 89.51 | 88.85 | 89.65 | 89.51 | 89.67 |
| $\Delta m\%$↑ | - | -0.91 | -0.04 | -0.94 | -0.33 | -1.71 | -0.22 | -1.04 | **0.04** | -1.10 | -0.13 | -0.90 | -0.04 | -0.12 | -0.04 |
| #P. | 85.62 | 42.81 | 85.61 | 42.81 | 43.43 | 42.81 | 43.43 | 42.81 | 43.43 | 42.81 | 43.43 | 42.81 | 43.43 | 85.62 | 105.70 |

Table 2: Multi-task learning results on CelebA dataset. All experiments are repeated over **3** random seeds and the mean values are reported. $\Delta m\%$ denotes the average relative improvement of all tasks. #P denotes model size (MB). The grey cell color indicates that Recon improves the result of the base model. The best average result is marked in bold.

| Method | Single-task | Joint-train | w/ Recon | CAGrad | w/ Recon | Graddrop | w/ Recon | PCGrad | w/ Recon |
|---|---|---|---|---|---|---|---|---|---|
| Average Error | 8.38 | 8.33 | 8.22 | 8.31 | 8.23 | 8.33 | **8.20** | 8.64 | 8.36 |
| $\Delta m\%$ ↑ | - | 0.55 | 1.92 | 0.79 | 1.74 | 0.23 | **2.13** | -3.14 | 0.24 |
| #P. | 1706.03 | 43.26 | 68.03 | 43.26 | 68.03 | 43.26 | 68.03 | 43.26 | 68.03 |

**Theorem 4.1.** *Assume that $\mathcal{L}$ is differentiable and for any two different tasks $\mathcal{T}_i$ and $\mathcal{T}_j$, it satisfies*

$$\cos \phi_{ij}^{(k)} \|\mathbf{g}_i^{(k)}\| < \|\mathbf{g}_j^{(k)}\|, \quad \forall k \in \mathbb{P} \tag{6}$$

*then for any sufficiently small learning rate $\alpha > 0$,*

$$\mathcal{L}(\hat{\theta}_r) < \mathcal{L}(\hat{\theta}). \tag{7}$$

The theorem indicates that a single gradient update on the model parameters of Recon achieves lower loss than that on the original model parameters. The proof is provided in Appendix A

## 5 EXPERIMENTS

In this section, we conduct extensive experiments to evaluate our approach Recon for multi-task learning and demonstrate its effectiveness, efficiency and generality.

### 5.1 EXPERIMENTAL SETUP

**Datasets.** We evaluate Recon on 4 multi-task datasets, namely **Multi-Fashion+MNIST** (Lin et al., 2019), **CityScapes** (Cordts et al., 2016), **NYUv2** (Couprie et al., 2013), **PASCAL-Context** (Mottaghi et al., 2014), and **CelebA** (Liu et al., 2015). The tasks of each dataset are described as follows. 1) Multi-Fashion+MNIST contains two image classification tasks. Each image consists of an item from FashionMNIST and an item from MNIST. 2) CityScapes contains 2 vision tasks: 7-class semantic segmentation and depth estimation. 3) NYUv2 contains 3 tasks: 13-class semantic segmentation, depth estimation and normal prediction. 4) PASCAL-Context consists of 5 tasks: semantic segmentation, human parts segmentation and saliency estimation, surface normal estimation, and edge detection. 5) CelebA contains 40 binary classification tasks.

**Baselines.** The baselines include 1) single-task learning (single-task): training all tasks independently; 2) joint-training (joint-train): training all tasks together with equal loss weights and all parameters shared; 3) gradient manipulation methods: MGDA (Sener & Koltun, 2018), PCGrad (Yu et al., 2020), GradDrop (Chen et al., 2020), CAGrad (Liu et al., 2021a), RotoGrad (Javaloy & Valera, 2022); 4) branched architecture search methods: BMTAS (Bruggemann et al., 2020); 5) Architecture design methods: Cross-Stitch (Misra et al., 2016), MMoE (Ma et al., 2018). Following Liu et al. (2021a), we implement Cross-Stitch based on SegNet (Badrinarayanan et al., 2017). For a fair comparison, all methods use same configurations and random seeds. We run all experiments 3 times with different random seeds. More experimental details are provided in Appendix B.

**Relative task improvement.** Following Maninis et al. (2019), we compute the relative task improvement with respect to the single-task baseline for each task. Given a task $\mathcal{T}_j$, the relative task

Table 3: Multi-task learning results on CityScapes dataset. All experiments are repeated over **3** random seeds and the mean values are reported. $\Delta m\%$ denotes the average relative improvement of all tasks. #P denotes the model size (MB). The grey cell color indicates that Recon improves the result of the base model. The best average result is marked in bold.

| Method | Segmentation (Higher Better) | | Depth (Lower Better) | | $\Delta m\% \uparrow$ | #P. |
|---|---|---|---|---|---|---|
| | mIoU | Pix Acc | Abs Err | Rel Err | | |
| Single-task | 74.36 | 93.22 | 0.0128 | 29.98 | | 190.59 |
| Cross-Stitch | 74.05 | 93.17 | 0.0162 | 116.66 | -79.04 | 190.59 |
| RotoGrad | 73.38 | 92.97 | 0.0147 | 82.31 | -47.81 | 103.43 |
| Joint-train | 74.13 | 93.13 | 0.0166 | 116.00 | -79.32 | 95.43 |
| w/ Recon | 74.17 | **93.21** | 0.0136 | 43.18 | -12.63 | 108.44 |
| MGDA | 70.74 | 92.19 | 0.0130 | 47.09 | -16.22 | 95.43 |
| w/ Recon | 71.01 | 92.17 | **0.0129** | **33.41** | **-4.46** | 108.44 |
| Graddrop | 74.08 | 93.08 | 0.0173 | 115.79 | -80.48 | 95.43 |
| w/ Recon | 74.17 | 93.11 | 0.0134 | 41.37 | -10.69 | 108.44 |
| PCGrad | 73.98 | 93.08 | 0.02 | 114.50 | -78.39 | 95.43 |
| w/ Recon | 74.18 | 93.14 | 0.0136 | 46.02 | -14.92 | 108.44 |
| CAGrad | 73.81 | 93.02 | 0.0153 | 88.29 | -53.81 | 95.43 |
| w/ Recon | 74.22 | 93.10 | 0.0130 | 38.27 | -7.38 | 108.44 |

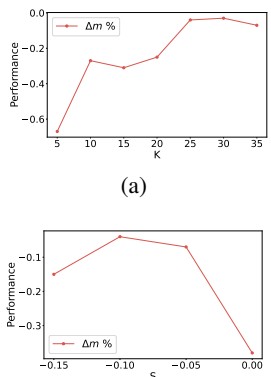

(a)

(b)

Figure 3: The performance of CAGrad combined with Recon on the Multi-Fashion+MNIST benchmark with (a) different number of selected layers $K$ (b) different severity value $S$ for computing conflict scores.

improvement is $\Delta m_{\mathcal{T}_j} = \frac{1}{K}\sum_{i=1}^{K}(-1)^{l_i}(M_i - S_i)/S_i$, where $M_i$, $S_i$ refer to metrics for the $i^{\text{th}}$ criterion obtained by objective model and single-task model respectively, $l_i = 1$ if a lower value for the criterion is better and 0 otherwise. The average relative task improvement is $\Delta m = \frac{1}{T}\sum_{j=1}^{T}\Delta m_{\mathcal{T}_j}$.

## 5.2 COMPARISON WITH THE STATE-OF-THE-ART

**Recon improves the performance of all base models.** The main results on Multi-Fashion+MNIST, and CelebA, CityScapes, PASCAL-Context, and NYUv2, are presented in Table 1, Table 2, Table 3, Table 4, and Table 5 respectively. **(1)** Compared to gradient manipulation methods, Recon consistently improves their performance in most evaluation metrics, and achieve comparable performance on the rest of evaluation metrics. **(2)** Compared with branched architecture search methods and architecture design methods, Recon can further improve the performance of BMTAS and MMoE. Besides, Recon combined with other gradient manipulation methods with small model size can achieve better results than branched architecture search methods with much bigger models.

**Small increases in model parameters can lead to good performance gains.** Note that Recon only changes a small portion of shared parameters to task-specific. As shown in Table 1-5, Recon increases the model size by 0.52% to 57.25%. Recon turns 1.42%, 1.46%, 12.77%, 0.26%, 9.80% shared parameters to task-specific on Multi-Fashion+MNIST, CelebA, CityScapes, NYUv2 and PASCAL-Context respectively. The results suggest that the gradient conflicts in a small portion (less than 13%) of shared parameters impede the training of the model for multi-task learning.

**Recon is compatible with various neural network architectures.** We use ResNet18 on Multi-Fashion+MNIST, SegNet (Badrinarayanan et al., 2017) on CityScapes, MTAN (Liu et al., 2019) on NYUv2, and MobileNetV2 (Sandler et al., 2018) on PASCAL-Context. Recon improves the performance of baselines with different neural network architectures, including the architecture search method BMTAS (Bruggemann et al., 2020) which finds a tree-like structure for multi-task learning.

**Only one search of conflict layers is needed for the same network architecture.** An interesting observation from our experiments is that network architecture seems to be the deciding factor for the conflict layers found by Recon. With the same network architecture (e.g., ResNet18), the found conflict layers are quite consistent w.r.t. (1) different training stages (e.g., the first 25% iterations, or the middle or last ones) (see Table 12 and Table 13 and discussion in Appendix C), (2) different MTL methods (e.g., joint-training or gradient manipulation methods) (see Table 14 and discussion in Appendix C), and (3) different datasets (see Table 15 and Table 16 and discussion in Appendix C).

Table 4: Multi-task learning results on PASCAL-Context dataset with 4-task setting. All experiments are repeated over **3** random seeds and the mean values are reported. $\Delta m\%$ denotes the average relative improvement of all tasks. #P denotes the model size (MB). The grey cell color indicates Recon improves the result of the base model. The best average result is marked in bold.

| Method | SemSeg (Higher Better) | | PartSeg (Lower Better) | | saliency (Higher Better) | Surface Normal Angle Distance (Lower Better) | | Within $t°$ (Higher Better) | | $\Delta m\% \uparrow$ | #P. |
|---|---|---|---|---|---|---|---|---|---|---|---|
| | mIoU | Pix Acc | mIoU | Pix Acc | mIoU | Mean | Median | 11.25 | 22.5 | | |
| Single-task | 65.00 | 90.53 | 59.59 | 92.61 | 65.61 | 14.55 | 12.36 | 46.51 | 81.29 | | 30.09 |
| Joint-train | 64.06 | 90.45 | 57.91 | 92.17 | 62.71 | 16.40 | 14.23 | 39.38 | 75.93 | -4.82 | 8.04 |
| w/ Recon | 64.73 | 90.50 | 59.00 | 92.44 | 66.17 | 14.99 | 12.68 | 44.82 | 80.11 | -0.66 | 10.20 |
| MGDA | 46.05 | 86.62 | 54.82 | 91.39 | 64.76 | 15.77 | 13.54 | 41.98 | 77.82 | -7.67 | 8.04 |
| w/ Recon | 55.82 | 87.73 | 56.31 | 91.67 | 64.91 | 15.12 | 12.88 | 44.36 | 79.81 | -4.14 | 10.20 |
| PCGrad | 63.91 | 90.45 | 58.01 | 92.19 | 63.09 | 16.34 | 14.19 | 39.62 | 76.06 | -4.59 | 8.04 |
| w/ Recon | 65.02 | 90.45 | 59.22 | 92.46 | 66.14 | 14.95 | 12.73 | 44.96 | 80.22 | -0.55 | 10.20 |
| Graddrop | 64.14 | 90.34 | 57.62 | 92.12 | 62.64 | 16.46 | 14.28 | 39.29 | 75.71 | -5.00 | 8.04 |
| w/ Recon | 64.48 | 90.45 | 59.08 | 92.46 | 66.23 | 14.94 | 12.72 | 45.03 | 80.25 | -0.63 | 10.20 |
| CAGrad | 63.37 | 90.17 | 57.49 | 92.07 | 64.16 | 16.30 | 14.12 | 39.80 | 76.23 | -4.37 | 8.04 |
| w/ Recon | 64.60 | 90.40 | 59.27 | 92.47 | 65.67 | 14.92 | 12.71 | 45.10 | 80.33 | -0.76 | 10.20 |
| BMTAS | 64.89 | 90.44 | 58.87 | 92.36 | 63.42 | 15.66 | 13.44 | 42.29 | 78.14 | -2.89 | 15.18 |
| w/ Recon | 64.78 | 90.46 | 59.96 | 92.58 | 65.96 | 14.74 | 12.57 | 45.62 | 80.84 | -0.19 | 16.83 |

Table 5: Multi-task learning results on NYUv2 dataset with MTAN as backbone. All experiments are repeated over **3** random seeds and the mean values are reported. $\Delta m\%$ denotes the average relative improvement of all tasks. #P denotes the model size (MB). The grey cell color indicates that Recon improves the result of the base model. The best average result is marked in bold.

| Method | Segmentation (Higher Better) | | Depth (Lower Better) | | Surface Normal Angle Distance (Lower Better) | | Within $t°$ (Higher Better) | | | $\Delta m\% \uparrow$ | #P. |
|---|---|---|---|---|---|---|---|---|---|---|---|
| | mIoU | Pix Acc | Abs Err | Rel Err | Mean | Median | 11.25 | 22.5 | 30 | | |
| Single-task | 38.67 | 64.27 | 0.6881 | 0.2788 | 24.87 | 18.99 | 30.43 | 57.81 | 69.70 | | 285.88 |
| Cross-Stitch | 40.45 | 66.15 | 0.5051 | 0.2134 | 27.58 | 23.00 | 24.69 | 49.47 | 62.36 | 4.16 | 285.88 |
| Joint-train | 39.48 | 65.23 | 0.5491 | 0.2235 | 27.87 | 23.76 | 22.68 | 47.91 | 61.58 | 0.75 | 168.72 |
| w/ Recon | 39.54 | 65.20 | 0.5312 | 0.2234 | 26.55 | 21.40 | 26.53 | 52.60 | 65.31 | 4.14 | 169.59 |
| MGDA | 29.28 | 60.30 | 0.6027 | 0.2515 | 24.89 | 19.32 | 29.85 | 57.18 | 69.38 | -2.26 | 168.72 |
| w/ Recon | 32.82 | 61.26 | 0.5884 | 0.2295 | 25.17 | 19.72 | 28.18 | 56.49 | 68.96 | 0.53 | 169.59 |
| Graddrop | 38.70 | 64.97 | 0.5565 | 0.2333 | 27.41 | 23.00 | 23.79 | 49.45 | 62.87 | 0.49 | 168.72 |
| w/ Recon | 40.14 | 66.08 | 0.5265 | 0.2241 | 26.51 | 21.45 | 26.51 | 52.48 | 65.26 | 4.67 | 169.59 |
| PCGrad | 38.55 | 65.07 | 0.54 | 0.23 | 26.90 | 22.05 | 24.98 | 51.36 | 64.41 | 2.02 | 168.72 |
| w/ Recon | 38.61 | 65.48 | 0.5350 | 0.2271 | 26.31 | 21.11 | 26.90 | 53.21 | 65.95 | 3.87 | 169.59 |
| CAGrad | 39.89 | 66.47 | 0.5496 | 0.2281 | 26.36 | 21.47 | 25.50 | 52.68 | 65.90 | 3.74 | 168.72 |
| w/ Recon | 39.92 | 66.07 | 0.5320 | 0.2200 | 25.80 | 20.59 | 27.60 | 54.31 | 67.05 | 5.80 | 169.59 |

Hence, in our experiments, we only search for the conflict layers *once* with the joint-training baseline in the first 25% training iterations and modify the network to improve various methods on the same dataset. We also find that the conflict layers found on one dataset can be used to modify the network to be directly applied on another dataset to gain performance improvement.

## 5.3 ABLATION STUDY AND ANALYSIS

**Recon greatly reduces the occurrence of conflicting gradients.** In Fig. 4 and Table 6, we compare the distribution of $\cos\phi_{ij}$ before and after applying Recon on Multi-Fashion+MNIST (the results on other datasets are provided in Appendix C). It can be seen that Recon greatly reduces the numbers of gradient pairs with severe conflicts ($\cos\phi_{ij} \in (-0.01, -1]$) by at least 67% and up to 79% when compared with joint-training, while gradient manipulation methods only slightly reduce the percentage and some even increases it. Similar observations can be made from Tables 8-10.

**Randomly selecting conflict layers does not work.** To show that the performance gain of Recon comes from selecting the layers with most severe conflicts instead of merely increasing model parameters, we further compare Recon with the following two baselines. RSL: randomly selecting

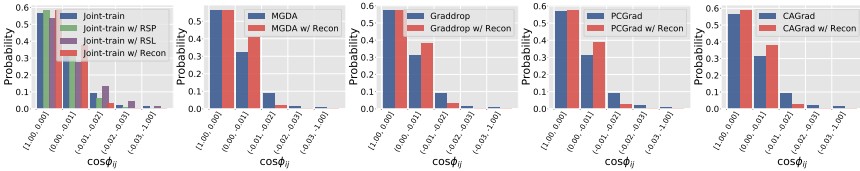

Figure 4: The distribution of gradient conflicts (in terms of $\cos \phi_{ij}$) of baselines and baselines with Recon on Multi-Fashion+MNIST dataset.

Table 6: The distribution of gradient conflicts (in terms of $\cos \phi_{ij}$) w.r.t. the shared parameters on Multi-Fashion+MNIST dataset. "Reduction" means the percentage of conflicting gradients in the interval of $(-0.01, -1.0]$ reduced by the model compared with joint-training. The grey cell color indicates Recon greatly reduces the conflicting gradients (more than 50%). In contrast, gradient manipulation methods only slightly decrease their occurrence, and some method even increases it.

| $\cos \phi_{ij}$ | Joint-train | w/ RSL | w/ RSP | w/ Recon | MGDA | w/ Recon | Graddrop | w/ Recon | PCGrad | w/ Recon | CAGrad | w/ Recon |
|---|---|---|---|---|---|---|---|---|---|---|---|---|
| [1.0, 0) | 56.56 | 53.44 | 58.15 | 58.53 | 56.06 | 56.50 | 57.26 | 57.61 | 56.72 | 57.75 | 56.18 | 59.06 |
| (0, -0.01] | 31.25 | 27.35 | 34.33 | 37.67 | 32.36 | 40.93 | 31.06 | 38.28 | 31.19 | 38.76 | 31.25 | 37.84 |
| (-0.01, -0.02] | 9.26 | 13.45 | 6.38 | 3.04 | 8.87 | 2.12 | 8.93 | 3.32 | 9.09 | 2.87 | 9.37 | 2.44 |
| (-0.02, -0.03] | 2.05 | 4.18 | 0.8 | 0.5 | 1.71 | 0.26 | 1.72 | 0.54 | 1.90 | 0.42 | 2.00 | 0.41 |
| (-0.03, -1.0] | 1.25 | 1.58 | 0.34 | 0.25 | 1.0 | 0.18 | 1.03 | 0.26 | 1.10 | 0.2 | 1.20 | 0.25 |
| Reduction (%) | - | -52.94 | 40.13 | 69.82 | 7.80 | 79.62 | 7.01 | 67.20 | 3.74 | 72.21 | -0.08 | 75.32 |

Table 7: Comparison of Recon with RSL and RSP. PD: performance drop compared to Recon.

| Seed | w/ RSL | w/ RSP | w/ Recon | CAGrad | | | | | PCGrad | | | | |
|---|---|---|---|---|---|---|---|---|---|---|---|---|---|
| | | | | Task 1 | | Task2 | | #P. | Task 1 | | Task2 | | #P. |
| | | | | Acc↑ | PD | Acc↑ | PD | | Acc↑ | PD | Acc↑ | PD | |
| 0 | ✓ | | | 97.60 | 0.68 | 64.39 | 25.26 | 73.02 | 97.43 | 0.87 | 65.57 | 24.21 | 73.02 |
| 1 | ✓ | | | 97.11 | 1.18 | 87.61 | 2.04 | 83.63 | 94.92 | 3.39 | 87.31 | 2.46 | 83.63 |
| 2 | ✓ | | | 94.62 | 3.66 | 87.68 | 1.96 | 76.33 | 92.90 | 5.40 | 87.41 | 2.36 | 76.33 |
| 0 | | ✓ | | 97.11 | 1.18 | 85.57 | 4.07 | 52.25 | 96.93 | 1.38 | 88.16 | 1.62 | 52.25 |
| 1 | | ✓ | | 97.81 | 0.47 | 88.28 | 1.36 | 51.96 | 97.63 | 0.68 | 88.55 | 1.22 | 51.96 |
| 2 | | ✓ | | 81.18 | 17.10 | 76.56 | 13.09 | 47.50 | 88.71 | 9.59 | 84.51 | 5.27 | 47.50 |
| - | - | - | ✓ | 98.28 | 0 | 89.65 | 0 | 43.42 | 98.30 | 0 | 89.77 | 0 | 43.42 |

same number of layers as Recon and set them task-specific. RSP: randomly selecting similar amount of parameters as Recon and set them task-specific. The results in Table 7 show that both RSL and RSP lead to significant performance drops, which verifies the effectiveness of the selection strategy of Recon. We compare Recon with the baselines that selects the first or last $K$ layers in Appendix C.

**Ablation study on hyperparameters.** We study the influence of the conflict severity $S$ and the number of selected layers $K$ on the performance of CAGrad w/ Recon on Multi-Fashion+MNIST. As shown in Fig. 3, a small $K$ leads to a significant performance drop, which indicates that there are still some shared network layers suffering from severe gradient conflicts, while a large $K$ will not lead to further performance improvement since severe conflicts have been resolved. For the conflict severity $S$, we find that a high value of $S$ (e.g., 0.0) leads to performance drops since it includes too many gradient pairs with small conflicts, while some of them are helpful for learning common structures and should not be removed. In the meantime, a too small $S$ (e.g., $-0.15$) also leads to performance degradation because it ignores too many gradient pairs with large conflicts, which may be detrimental to learning. While $K$ and $S$ are sensitive, we may only need to tune them once for a given network architecture, as discussed in Sec. 5.2.

## 6 CONCLUSION

We have proposed a very simple yet effective approach, namely Recon, to reduce the occurrence of conflicting gradients for multi-task learning. By considering layer-wise gradient conflicts and identifying the shared layers with severe conflicts and setting them task-specific, Recon can significantly reduce the occurrence of severe conflicting gradients and boost the performance of existing methods with only a reasonable increase in model parameters. We have demonstrated the effectiveness, efficiency, and generality of Recon via extensive experiments and analysis.

ACKNOWLEDGMENTS

The authors would like to thank Lingzi Jin for checking the proof of Theorem A.1 and the anonymous reviewers for their insightful and helpful comments.

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

# A PROOF OF THEOREM A.1

**Theorem A.1.** *Assume that $\mathcal{L}$ is differentiable and for any two different tasks $\mathcal{T}_i$ and $\mathcal{T}_j$, it satisfies*

$$\cos \phi_{ij}^{(k)} \|\mathbf{g}_i^{(k)}\| < \|\mathbf{g}_j^{(k)}\|, \quad \forall k \in \mathbb{P} \tag{8}$$

*then for any sufficiently small learning rate $\alpha > 0$,*

$$\mathcal{L}(\hat{\theta}_r) < \mathcal{L}(\hat{\theta}). \tag{9}$$

*Proof.* We consider the first order Taylor approximation of $\mathcal{L}_i$. For normal update, we have

$$\mathcal{L}_i \left( \hat{\theta}_{\text{sh}}^{\text{fix}}, \hat{\theta}_{\text{sh}}^{\text{cf}}, \hat{\theta}_i^{\text{ts}} \right) = \mathcal{L}_i \left( \theta_{\text{sh}}^{\text{fix}}, \theta_{\text{sh}}^{\text{cf}}, \theta_i^{\text{ts}} \right) + (\hat{\theta}_{\text{sh}}^{\text{fix}} - \theta_{\text{sh}}^{\text{fix}})^\top \mathbf{g}_i^{\text{fix}} \tag{10}$$

$$+ (\hat{\theta}_{\text{sh}}^{\text{cf}} - \theta_{\text{sh}}^{\text{cf}})^\top \mathbf{g}_i^{\text{cf}} + (\hat{\theta}_i^{\text{ts}} - \theta_i^{\text{ts}})^\top \mathbf{g}_i^{\text{ts}} + o(\alpha). \tag{11}$$

For Recon update, we have

$$\mathcal{L}_i \left( \hat{\theta}_{\text{sh}}^{\text{fix}}, \hat{\theta}_i^{\text{cf}}, \hat{\theta}_i^{\text{ts}} \right) = \mathcal{L}_i \left( \theta_{\text{sh}}^{\text{fix}}, \theta_{\text{sh}}^{\text{cf}}, \theta_i^{\text{ts}} \right) + (\hat{\theta}_{\text{sh}}^{\text{fix}} - \theta_{\text{sh}}^{\text{fix}})^\top \mathbf{g}_i^{\text{ts}} \tag{12}$$

$$+ (\hat{\theta}_i^{\text{cf}} - \theta_{\text{sh}}^{\text{cf}})^\top \mathbf{g}_i^{\text{cf}} + (\hat{\theta}_i^{\text{ts}} - \theta_i^{\text{ts}})^\top \mathbf{g}_i^{\text{ts}} + o(\alpha). \tag{13}$$

The difference between the two loss functions after the update is

$$\mathcal{L}_i \left( \hat{\theta}_{\text{sh}}^{\text{fix}}, \hat{\theta}_i^{\text{cf}}, \hat{\theta}_i^{\text{ts}} \right) - \mathcal{L}_i \left( \hat{\theta}_{\text{sh}}^{\text{fix}}, \hat{\theta}_{\text{sh}}^{\text{cf}}, \hat{\theta}_i^{\text{ts}} \right) = (\hat{\theta}_i^{\text{cf}} - \hat{\theta}_{\text{sh}}^{\text{cf}})^\top \mathbf{g}_i^{\text{cf}} + o(\alpha) \tag{14}$$

$$= -\alpha \left( \mathbf{g}_i^{\text{cf}} - \sum_{j=1}^T w_j \mathbf{g}_j^{\text{cf}} \right)^\top \mathbf{g}_i^{\text{cf}} + o(\alpha) \tag{15}$$

$$= -\alpha \sum_{j=1}^T w_j \left( \mathbf{g}_i^{\text{cf}} - \mathbf{g}_j^{\text{cf}} \right)^\top \mathbf{g}_i^{\text{cf}} + o(\alpha) \tag{16}$$

$$= -\alpha \sum_{j=1}^T w_j \left( \|\mathbf{g}_i^{\text{cf}}\|^2 - \mathbf{g}_j^{\text{cf}\top} \mathbf{g}_i^{\text{cf}} \right) + o(\alpha). \tag{17}$$

Assume, without loss of generality, that $\|\mathbf{g}_i^{cf}\| \neq 0$, then

$$\left\| \mathbf{g}_i^{\text{cf}} \right\|^2 - \mathbf{g}_j^{\text{cf}\top} \mathbf{g}_i^{\text{cf}} = \sum_{k \in \mathbb{P}} \left( \left\| \mathbf{g}_i^{(k)} \right\|^2 - \mathbf{g}_i^{(k)\top} \mathbf{g}_j^{(k)} \right) \tag{18}$$

$$= \sum_{k \in \mathbb{P}} \left\| \mathbf{g}_i^{(k)} \right\| \left( \left\| \mathbf{g}_i^{(k)} \right\| - \cos \phi_{ij}^{(k)} \left\| \mathbf{g}_j^{(k)} \right\| \right) \tag{19}$$

$$> 0. \tag{20}$$

Hence, the above difference is negative, if $\alpha$ is sufficiently small. As such, the difference between the multi-task loss functions is also negative, if $\alpha$ is sufficiently small.

$$\mathcal{L}(\hat{\theta}_r) - \mathcal{L}(\hat{\theta}) = \sum_{i=1}^T \mathcal{L}_i \left( \hat{\theta}_{\text{sh}}^{\text{fix}}, \hat{\theta}_i^{\text{cf}}, \hat{\theta}_i^{\text{ts}} \right) - \sum_{i=1}^T \mathcal{L}_i \left( \hat{\theta}_{\text{sh}}^{\text{fix}}, \hat{\theta}_{\text{sh}}^{\text{cf}}, \hat{\theta}_i^{\text{ts}} \right) < 0 \tag{21}$$

$\square$

# B EXPERIMENTAL SETUP

## B.1 MULTI-FASHION+MNIST

**Model.** We adopt ResNet18 (He et al., 2016) without pre-training as the backbone and modify the dimension of the output features to 100 for the last linear layer. For the task-specific heads, we define two linear layers followed by a ReLU function.

**Tasks, losses, and metrics.** Each task is a classification problem with 10 classes and we use the cross-entropy loss as the classification loss. For evaluation, we use the classification accuracy as the metric for each task.

**Model hyperparameters.** We train the model for 120 epochs with the batch size of 256. We adopt SGD with an initial learning rate of 0.1 and decay the learning rate by 0.1 at the 60th and 90th epoch.

**Baseline hyperparameters.** For CAGrad, we set $\alpha = 0.2$. For BMTAS, we set the resource loss weight to 1.0, and we search the architecture for 100 epochs. For RotoGrad, we set $R_k = 100$ which is equal to the dimension of shared features and set the learning rate of rotation parameters as learning rate of the neural networks. For MMoE, the initial learning rate of expert networks and gates are 0.1 and $1e - 3$ respectively.

**Recon hyperparameters.** We use CAGrad to train the model for 30 epochs and compute the conflict score of each shared layer. We set $S = -0.1$ for computing the scores. We select 25 layers with the highest conflict scores and turn them into task-specific layers.

## B.2 CITYSCAPES

**Model.** We adopt SegNet (Badrinarayanan et al., 2017) as the backbone where the decoder is split into two convolutional heads.

**Model hyperparameters.** We train the model for 200 epochs with the batch size of 8. We adopt Adam with an initial learning rate of $5e - 5$ and decay the learning rate by 0.5 at the 100th epoch.

**Baselines hyperparameters.** For CAGrad, we set $\alpha = 0.2$. For RotoGrad, we set $R_k = 1024$ and set the learning rate of rotation parameters as 10 times less than the learning rate of the neural networks.

**Recon hyperparameters.** We use joint-train to train the model for 40 epochs and compute the conflict score of each shared layer. We set $S = 0.0$ for computing the scores. We select 39 layers with the highest conflict scores and turn them into task-specific layers.

## B.3 NYUV2

**Model.** We adopt MTAN (Liu et al., 2019) – the SegNet combined with task-specific attention modules on the encoder.

**Model hyperparameters.** We train the model for 200 epochs with the batch size of 2. We adopt Adam with an initial learning rate of $1e - 4$ and decay the learning rate by 0.5 at the 100th epoch.

**Baseline hyperparameters.** For CAGrad, we set $\alpha = 0.4$ similar with Liu et al. (2021a).

**Recon hyperparameters.** We use joint-train to train the model for 40 epochs and compute the conflict score of each shared layer. We set $S = -0.02$ for computing the scores. We select 22 layers with the highest conflict scores and turn them into task-specific layers.

## B.4 PASCAL-CONTEXT

**Model.** Following Bruggemann et al. (2020), we employ MobileNetv2 Sandler et al. (2018) as the backbone with a reduced design of the ASPP module (R-ASPP) (Sandler et al., 2018). We pre-train the model on ImageNet (Deng et al., 2009).

**Model hyperparameters.** We train the model for 130 epochs with the batch size of 6. We adopt Adam with an initial learning rate of $1e - 4$ and decay the learning rate by 0.1 at the 70th and 100th epoch.

**Baselines hyperparameters.** For CAGrad, we set $\alpha = 0.1$. For BMTAS, we set the resoure loss weight to 0.1, and we search the architecture for 130 epochs.

**Recon hyperparameters.** We use joint-train to train the model for 40 epochs and compute the conflict score of each shared layer. We set $S = -0.02$ for computing the scores. We select 85 layers with the highest conflict scores and turn them into task-specific layers.

### B.5 CELEBA

**Model.** Following Sener & Koltun (2018), we use ResNet18 (He et al., 2016) as the backbone network. We pre-train the model on ImageNet (Deng et al., 2009).

**Model hyperparameters.** We train the model for 5 epochs. We adopt Adam with an initial learning rate of $5e-5$ and decay the learning rate by $0.5$ at the $3^{\text{th}}$ epoch.

**Baselines hyperparameters.** For CAGrad, we set $\alpha = 0.1$.

**Recon hyperparameters.** We use joint-train to train the model for 2 epochs and compute the conflict score of each shared layer. We set $S = -0.05$. We select 25 layers with the highest conflict scores and turn them into task-specific layers.

## C    ADDITIONAL ABLATION STUDY

**The distribution of gradient conflicts**. In addition to the statistics on Multi-Fashion+MNIST, we further show the distributions of gradient conflicts of various baselines on CityScapes, NYUv2, and PASCAL-Context in Fig 5, Fig 6, and Fig 7 respectively. We compare the distributions with those of baselines w/ Recon on the three datasets in Fig. 8, Fig. 9, and Fig. 10 respectively. The detailed statistics are provided in Tables 8-10.

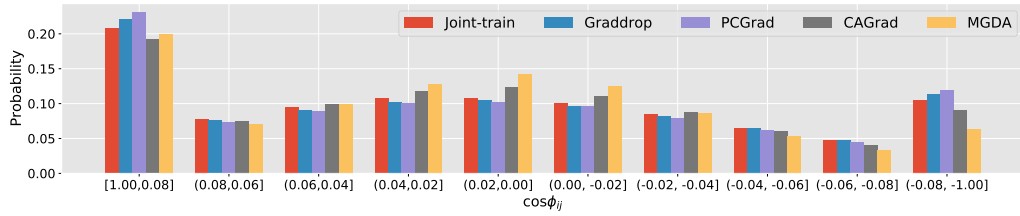

Figure 5: The distributions of gradient conflicts (in terms of $\cos \phi_{ij}$) of the joint-training baseline and state-of-the-art gradient manipulation methods on CityScapes dataset.

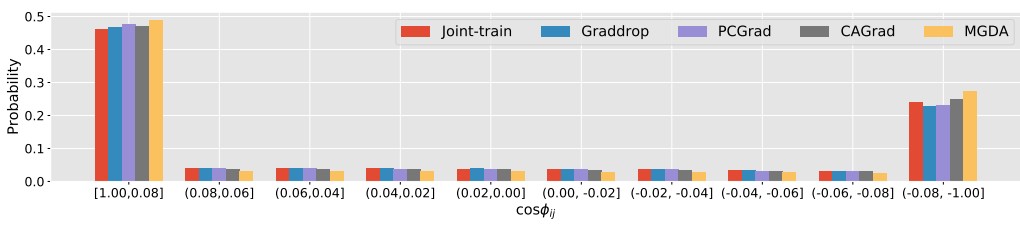

Figure 6: The distributions of gradient conflicts (in terms of $\cos \phi_{ij}$) of the joint-training baseline and state-of-the-art gradient manipulation methods on NYUv2 dataset.

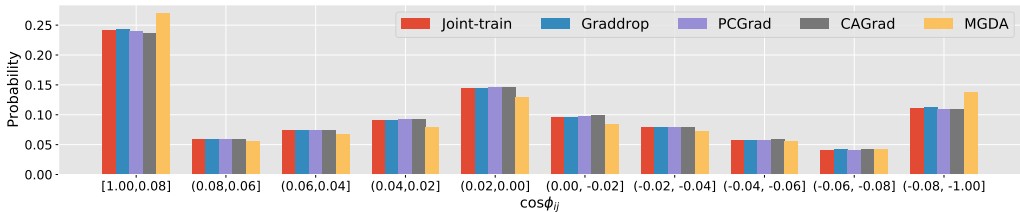

Figure 7: The distributions of gradient conflicts (in terms of $\cos \phi_{ij}$) of the joint-training baseline and state-of-the-art gradient manipulation methods on PASCAL-Context dataset.

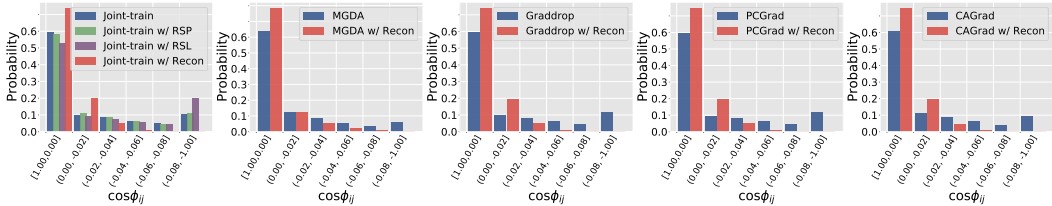

Figure 8: The distribution of gradient conflicts (in terms of $\cos \phi_{ij}$) w.r.t. the shared parameters on CityScapes. RSL: randomly selecting same number of layers as Recon and set them task-specific. RSP: randomly selecting similar amount of parameters as Recon and set them task-specific.

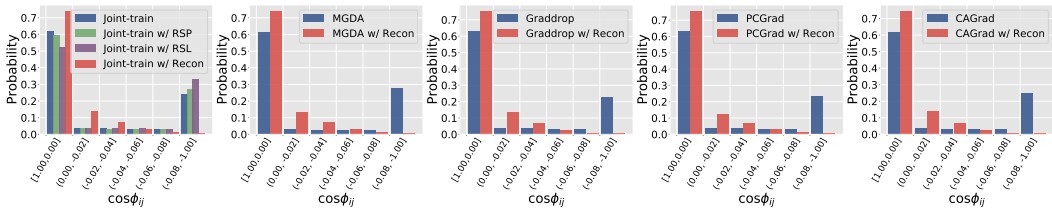

Figure 9: The distribution of gradient conflicts (in terms of $\cos \phi_{ij}$) of baselines and baselines with Recon on NYUv2. RSL: randomly selecting same number of layers as Recon and set them task-specific. RSP: randomly selecting similar amount of parameters as Recon and set them task-specific.

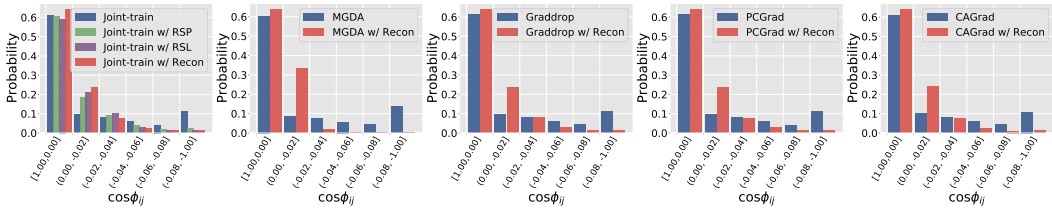

Figure 10: The distribution of gradient conflicts (in terms of $\cos \phi_{ij}$) of baselines and baselines with Recon on PASCAL-Context. RSL: randomly selecting same number of layers as Recon and set them task-specific. RSP: randomly selecting similar amount of parameters as Recon and set them task-specific.

Table 8: The distribution of gradient conflicts (in terms of $\cos \phi_{ij}$) w.r.t. the shared parameters on CityScapes dataset. "Reduction" means the percentage of conflicting gradients in the interval of $(-0.02, -1.0]$ reduced by the model compared with joint-training. The grey cell color indicates Recon greatly reduces the conflicting gradients (more than 50%). In contrast, gradient manipulation methods only moderately decrease their occurrence (MGDA deceases it by 22%), and some methods even increase it.

| $\cos \phi_{ij}$ | Joint-train | w/ RSL | w/ RSP | w/ Recon | MGDA | w/ Recon | Graddrop | w/ Recon | PCGrad | w/ Recon | CAGrad | w/ Recon |
|---|---|---|---|---|---|---|---|---|---|---|---|---|
| [1.0, 0) | 59.55 | 53.16 | 58.29 | 73.62 | 63.9 | 78.27 | 59.56 | 73.82 | 59.85 | 74.52 | 60.79 | 74.54 |
| (0, -0.02] | 10.14 | 9.01 | 10.77 | 20.13 | 12.51 | 12.54 | 9.61 | 19.75 | 9.58 | 19.43 | 11.13 | 19.77 |
| (-0.02, -0.04] | 8.52 | 7.34 | 8.72 | 5.13 | 8.59 | 5.54 | 8.19 | 5.17 | 7.94 | 4.89 | 8.83 | 4.62 |
| (-0.04, -0.06] | 6.45 | 5.69 | 6.48 | 0.94 | 5.39 | 2.23 | 6.49 | 1.05 | 6.24 | 0.96 | 6.05 | 0.89 |
| (-0.06, -0.08] | 4.79 | 4.53 | 4.61 | 0.14 | 3.29 | 0.85 | 4.76 | 0.16 | 4.41 | 0.15 | 4.06 | 0.13 |
| (-0.08, -1.0] | 10.54 | 20.26 | 11.13 | 0.03 | 6.33 | 0.56 | 11.38 | 0.05 | 11.98 | 0.06 | 9.13 | 0.04 |
| Reduction (%) | - | -24.82 | -2.11 | 79.41 | 22.11 | 69.70 | -1.72 | 78.78 | -0.89 | 80.03 | 7.36 | 81.22 |

Table 9: The distribution of gradient conflicts (in terms of $\cos \phi_{ij}$) w.r.t. the shared parameters on NYUv2 dataset. "Reduction" means the percentage of conflicting gradients in the interval of $(-0.04, -1.0]$ reduced by the model compared with joint-training. The grey cell color indicates Recon greatly reduces the conflicting gradients (more than 50%). In contrast, gradient manipulation methods only slightly decrease their occurrence, and some methods even increase it.

| $\cos \phi_{ij}$ | Joint-train | w/ RSL | w/ RSP | w/ Recon | MGDA | w/ Recon | Graddrop | w/ Recon | PCGrad | w/ Recon | CAGrad | w/ Recon |
|---|---|---|---|---|---|---|---|---|---|---|---|---|
| [1.0, 0) | 61.96 | 52.61 | 59.70 | 73.99 | 61.28 | 74.08 | 62.93 | 75.35 | 63.25 | 75.54 | 61.95 | 74.49 |
| (0, -0.02] | 3.85 | 3.75 | 3.47 | 14.17 | 2.97 | 13.38 | 3.83 | 13.50 | 3.61 | 12.66 | 3.53 | 14.20 |
| (-0.02, -0.04] | 3.63 | 3.60 | 3.41 | 7.07 | 2.77 | 7.21 | 3.70 | 6.71 | 3.62 | 6.66 | 3.39 | 6.96 |
| (-0.04, -0.06] | 3.39 | 3.43 | 3.11 | 2.89 | 2.81 | 3.19 | 3.45 | 2.71 | 3.26 | 2.98 | 3.21 | 2.71 |
| (-0.06, -0.08] | 3.11 | 3.30 | 2.94 | 1.13 | 2.64 | 1.28 | 3.16 | 1.03 | 3.06 | 1.25 | 3.05 | 1.01 |
| (-0.08, -1.0] | 24.05 | 33.31 | 27.37 | 0.76 | 27.53 | 0.87 | 22.92 | 0.70 | 23.20 | 0.90 | 24.88 | 0.63 |
| Reduction (%) | - | -31.06 | -9.39 | 84.35 | -7.95 | 82.52 | 3.34 | 85.47 | 3.37 | 83.21 | -1.93 | 85.76 |

Table 10: The distribution of gradient conflicts (in terms of $\cos \phi_{ij}$) w.r.t. the shared parameters on PASCAL-Context dataset. "Reduction" means the percentage of conflicting gradients in the interval of $(-0.02, -1.0]$ reduced by the model compared with joint-training. The grey cell color indicates Recon greatly reduces the conflicting gradients (more than 50%). In contrast, gradient manipulation methods only slightly decrease their occurrence, and some methods even increase it.

| $\cos \phi_{ij}$ | Joint-train | w/ RSL | w/ RSP | w/ Recon | MGDA | w/ Recon | Graddrop | w/ Recon | PCGrad | w/ Recon | CAGrad | w/ Recon |
|---|---|---|---|---|---|---|---|---|---|---|---|---|
| [1.0, 0) | 61.26 | 59.20 | 60.47 | 63.99 | 60.40 | 63.61 | 61.18 | 63.76 | 61.35 | 63.83 | 60.99 | 63.78 |
| (0, -0.02] | 9.66 | 21.01 | 18.25 | 23.57 | 8.51 | 33.53 | 9.66 | 23.41 | 9.83 | 23.61 | 9.95 | 24.04 |
| (-0.02, -0.04] | 7.90 | 9.91 | 9.10 | 7.65 | 7.27 | 2.04 | 7.89 | 7.83 | 7.90 | 7.65 | 8.03 | 7.53 |
| (-0.04, -0.06] | 5.85 | 3.05 | 3.88 | 2.59 | 5.68 | 0.45 | 5.80 | 2.71 | 5.82 | 2.66 | 5.91 | 2.51 |
| (-0.06, -0.08] | 4.16 | 1.32 | 1.79 | 1.07 | 4.35 | 0.17 | 4.21 | 1.12 | 4.13 | 1.10 | 4.23 | 1.04 |
| (-0.08, -1.0] | 11.16 | 1.30 | 2.29 | 1.13 | 13.80 | 0.20 | 11.24 | 1.16 | 10.97 | 1.16 | 10.88 | 1.08 |
| Reduction (%) | - | 46.41 | 41.31 | 57.21 | -6.98 | 90.16 | -0.24 | 55.90 | 0.86 | 56.76 | 0.07 | 58.20 |

Table 11: Multi-task learning results on Multi-Fashion+MNIST dataset. LSK refers to turning the fist $K$ layers into task-specific layers. FSK refers to turning the last $K$ layers into task-specific layers. PD denotes the performance drop compared with Recon.

| | | | CAGrad | | | | | PCGrad | | | | |
|---|---|---|---|---|---|---|---|---|---|---|---|---|
| LSK | FSK | w/ Recon | Task 1 | | Task2 | | #P. | Task 1 | | Task2 | | #P. |
| | | | Acc↑ | PD | Acc↑ | PD | | Acc↑ | PD | Acc↑ | PD | |
| ✓ | | | 97.63 | **0.66** | 89.14 | **0.50** | 84.17 | 97.63 | **0.65** | 88.98 | **0.66** | 84.17 |
| | ✓ | | 98.21 | **0.07** | 89.15 | **0.50** | 48.90 | 98.19 | **0.09** | 89.51 | **0.13** | 48.90 |
| - | | ✓ | 98.28 | 0 | 89.65 | 0 | 43.42 | 98.30 | 0 | 89.77 | 0 | 43.42 |

**Selecting the first $K$ layers and the last $K$ Layers as conflict layers does not work.** To further support the conclusion that the selection of parameters with higher probability of conflicting gradients contributes most to the performance gain rather than the increase in model capacity. We compare Recon with two baselines: (1) Select the first $K$ neural network layers and turn them into task-specific layers. (2) Select the last $K$ neural network layers and turn them into task-specific layers. The multi-task learning results on the Multi-Fashion+MNIST benchmark are presented in Table 11. The results show that if we directly turn the top or the bottom of the neural network into task-specific parameters, it still will lead to performance degradation compared to Recon.

**Recon finds similar layers in different training stages.** Recon ranks the network layers according to the computed $S$-conflict scores. The ranking result can be represented as a layer permutation, denoted as $\pi$, and $\pi(l)$ is the position of layer $l$. The similarity between two rankings $\pi_i$ and $\pi_j$ can be measured as:

$$d(\pi_i, \pi_j) = \frac{1}{|\mathbb{L}|} \sum_{l \in \mathbb{L}} |\pi_i(l) - \pi_j(l)|, \quad (22)$$

where $\mathbb{L}$ denotes the set of neural network layers. In Table 12, we measure the differences in rankings obtained in different training stages (e.g., in the first 25% iterations or the second 25% iterations)

Table 12: The distance between the layer permutations (rankings) obtained in different training stages on Multi-Fashion+MNIST dataset. "Iter." denotes iterations.

| Training Stage | 1st 25% Iter. | 2nd 25% Iter. | 3rd 25% Iter. | 4th 25% Iter. | All Iter. |
|---|---|---|---|---|---|
| 1st 25% Iter. | 0 | - | - | - | - |
| 2nd 25% Iter. | **2.39** | 0 | - | - | - |
| 3rd 25% Iter. | 1.85 | 2.14 | 0 | - | - |
| 4th 25% Iter. | 1.95 | 2.24 | 0.68 | 0 | - |
| All Iter. | 1.36 | 1.95 | 0.82 | 0.97 | 0 |

Table 13: Performance of the networks modified by Recon with conflict layers found in different training stages of joint-training on CityScapes dataset. $\Delta m\%$ denotes the average relative improvement of all tasks. #P denotes the model size (MB). The best result is marked in bold.

| | Segmentation | | Depth | | | |
| Model | (Higher Better) | | (Lower Better) | | $\Delta m\%$ | #P. |
| | mIoU | Pix Acc | Abs Err | Rel Err | | |
|---|---|---|---|---|---|---|
| Single-task | 74.36 | 93.22 | 0.0128 | 29.98 | | 190.59 |
| 1st 25% Iterations | 74.17 | 93.21 | 0.0136 | 43.18 | -12.63 | 108.439 |
| 2nd 25% Iterations | 74.20 | 93.19 | 0.0135 | 42.45 | -11.83 | 108.440 |
| 3rd 25% Iterations | **74.80** | 93.19 | **0.0136** | **41.34** | -10.90 | 109.567 |
| 4th 25% Iterations | **74.80** | 93.19 | **0.0136** | **41.34** | -10.90 | 109.567 |
| All Iterations | **74.80** | 93.19 | **0.0136** | **41.34** | -10.90 | 109.567 |

on Multi-Fashion+MNIST by Eq. 22. The small distances (less than 2.4) indicate that the layers found in different training stages are quite similar. In Table 13, we compare the performance of the networks modified by Recon with conflict layers found in different training stages on CityScapes. It can be seen that the results of the last three rows are the same, which is because the layers found in the 3rd 25% iterations, 4th 25% iterations, and all iterations are *exactly the same* (the rankings may be slightly different though). The layers found in the later stages lead to slightly better performance than those found in the early stages (i.e., 1st 25% iterations and 2nd 25% iterations), indicating the conflict scores in early iterations might be a little noisy. However, since the performance gaps are acceptably small, to save time, we use the initial 25% training iterations to find conflict layers.

Table 14: The distance between the layer permutations (rankings) obtained by Recon with different methods on Multi-Fashion+MNIST dataset.

| Method | Joint-train | CAGrad | PCGrad | Gradrop | MGDA |
|---|---|---|---|---|---|
| Joint-train | 0 | - | - | - | - |
| CAGrad | 1.07 | 0 | - | - | - |
| PCGrad | 0.78 | 1.17 | 0 | - | - |
| Gradrop | 0.59 | 0.83 | 0.68 | 0 | - |
| MGDA | 1.71 | 1.32 | **1.90** | 1.56 | 0 |

**Recon finds similar layers with different MTL methods.** In Table 14, we measure the differences in layer permutations (rankings) obtained by Recon with different methods (e.g., CAGrad and PC-Grad) on Multi-Fashion+MNIST by Eq. 22. The small distances (less than 1.9) indicate that the layers found by Recon with different methods are quite similar. Therefore, in our experiments, we only use joint-training to search for the conflict layers once, and directly apply the modified network to improve different gradient manipulation methods as shown in Tables 1-5.

**The conflict layers found by Recon with the same architecture are transferable between different datasets.** We conduct experiments with three different architectures: ResNet18, SegNet, and MTAN. **(1)** For Resnet18, we find that the layers found by Recon on CelebA and those found on Multi-Fashion+MNIST are *exactly the same*. **(2)** For SegNet, we find that 95% layers (38 out of 40)

Table 15: Multi-task learning results on NYUv2 dataset with SegNet as backbone. Recon* denotes setting the layers found on CityScapes to task-specific. $\Delta m\%$ denotes the average relative improvement of all tasks. #P denotes the model size (MB). The grey cell color indicates that Recon or Recon* improves the result of the base model.

| Method | Segmentation (Higher Better) | | Depth (Lower Better) | | Surface Normal | | | | | $\Delta m\%\uparrow$ | #P. |
| | | | | | Angle Distance (Lower Better) | | Within $t°$ (Higher Better) | | | | |
| | mIoU | Pix Acc | Abs Err | Rel Err | Mean | Median | 11.25 | 22.5 | 30 | | |
| Single-task | 38.67 | 64.27 | 0.6881 | 0.2788 | 24.8683 | 18.9919 | 30.43 | 57.81 | 69.7 | | 285.88 |
| Joint-train | 38.62 | 65.36 | 0.5378 | 0.2273 | 29.92 | 25.82 | 20.79 | 44.29 | 57.36 | -1.62 | 95.58 |
| w/ Recon | 40.68 | 66.12 | 0.5786 | 0.2558 | 26.72 | 21.41 | 26.58 | 52.58 | 65.20 | 2.15 | 139.59 |
| w/ Recon* | 38.81 | 63.69 | 0.5637 | 0.2413 | 26.75 | 21.73 | 26.16 | 51.80 | 64.64 | 1.59 | 121.59 |
| MGDA | 25.71 | 57.72 | 0.6033 | 0.2358 | 24.53 | 18.65 | 31.22 | 58.46 | 70.21 | -2.15 | 95.58 |
| w/ Recon | 36.64 | 62.36 | 0.5613 | 0.2255 | 24.66 | 18.66 | 31.30 | 58.47 | 70.16 | 5.37 | 139.59 |
| w/ Recon* | 36.85 | 63.51 | 0.5760 | 0.2362 | 24.89 | 18.96 | 30.53 | 57.94 | 69.82 | 4.34 | 121.59 |
| Graddrop | 39.01 | 66.13 | 0.5462 | 0.2296 | 29.72 | 25.51 | 19.87 | 44.68 | 58.12 | -1.52 | 95.58 |
| w/ Recon | 39.78 | 65.63 | 0.5460 | 0.2280 | 26.42 | 21.16 | 26.89 | 53.16 | 65.84 | 4.45 | 139.59 |
| w/ Recon* | 39.97 | 65.71 | 0.5544 | 0.2261 | 26.52 | 21.37 | 26.65 | 52.65 | 65.46 | 4.21 | 121.59 |
| PCGrad | 40.01 | 65.77 | 0.5349 | 0.2227 | 28.53 | 24.08 | 22.33 | 47.42 | 60.69 | 1.43 | 95.58 |
| w/ Recon | 40.03 | 65.92 | 0.5523 | 0.2384 | 26.24 | 20.89 | 27.30 | 53.66 | 66.25 | 4.19 | 139.59 |
| w/ Recon* | 39.93 | 65.46 | 0.5494 | 0.2315 | 26.82 | 21.70 | 26.34 | 52.04 | 64.74 | 3.53 | 121.59 |
| CAGrad | 38.87 | 66.54 | 0.5331 | 0.2289 | 25.85 | 20.60 | 27.50 | 54.41 | 67.10 | 5.60 | 95.58 |
| w/ Recon | 40.68 | 66.12 | 0.5372 | 0.2266 | 25.44 | 19.87 | 28.96 | 56.00 | 68.28 | 6.99 | 139.59 |
| w/ Recon* | 39.97 | 65.92 | 0.5298 | 0.2273 | 25.56 | 20.11 | 28.69 | 55.37 | 67.75 | 6.47 | 121.59 |

Table 16: Multi-task learning results on CityScapes dataset with MTAN as backbone. Recon* denotes setting the layers found on NYUv2 to task-specific. $\Delta m\%$ denotes the average relative improvement of all tasks. #P denotes the model size (MB). The grey cell color indicates that Recon or Recon* improves the result of the base model.

| Method | Segmentation (Higher Better) | | Depth (Lower Better) | | $\Delta m\%\uparrow$ | #P. |
| | mIoU | Pix Acc | Abs Err | Rel Err | | |
| Single-task | 73.74 | 93.05 | 0.0129 | 27.71 | | 190.58 |
| Joint-train | 75.35 | 93.55 | 0.0169 | 45.64 | -23.26 | 157.19 |
| w/ Recon | 75.72 | 93.74 | 0.0130 | 40.90 | -11.36 | 196.32 |
| w/ Recon* | 76.32 | 93.76 | 0.0132 | 46.40 | -16.44 | 159.19 |
| MGDA | 70.46 | 91.75 | 0.0224 | 34.33 | -26.02 | 157.19 |
| w/ Recon | 72.23 | 92.60 | **0.0122** | 26.93 | **1.37** | 196.32 |
| w/ Recon* | 70.83 | 92.14 | 0.0125 | **25.69** | 1.31 | 159.19 |
| Graddrop | 75.19 | 93.53 | 0.0168 | 46.35 | -23.90 | 157.19 |
| w/ Recon | 75.60 | 93.72 | 0.0127 | 38.55 | -8.71 | 196.32 |
| w/ Recon* | **76.49** | **93.82** | 0.0129 | 47.54 | -16.81 | 159.19 |
| PCGrad | 75.64 | 93.54 | 0.02 | 43.53 | -23.60 | 157.19 |
| w/ Recon | 75.89 | 93.71 | 0.0129 | 40.05 | -10.35 | 196.32 |
| w/ Recon* | 76.24 | 93.69 | 0.0128 | 45.24 | -14.66 | 159.19 |
| CAGrad | 75.26 | 93.50 | 0.0176 | 44.23 | -23.40 | 157.19 |
| w/ Recon | 75.65 | 93.71 | 0.0125 | 36.23 | -6.15 | 196.32 |
| w/ Recon* | 76.25 | 93.74 | 0.0123 | 40.05 | -8.99 | 159.19 |

found on NYUv2 are identical to those found on CityScapes. On NYUv2, we compare the performance of using conflict layers found on NYUv2 (baselines w/ Recon) to that of using conflict layers found on CityScapes (i.e., baselines w/ Recon*), as shown in Table 15. **(3)** For MTAN (SegNet with attention), we find that 68% layers (17 out of 25) found on CityScapes are identical to those found on NYUv2. On CityScapes, we compare the performance of using conflict layers found on CityScapes (baselines w/ Recon) to that of using conflict layers found on NYUv2 (i.e., baselines w/ Recon*), as shown in Table 16. The results show that the conflict layers found on one dataset can be used to modify the network to be directly used on another dataset to consistently improve the performance of various baselines, while searching for the conflict layers again on the new dataset may lead to better performance.

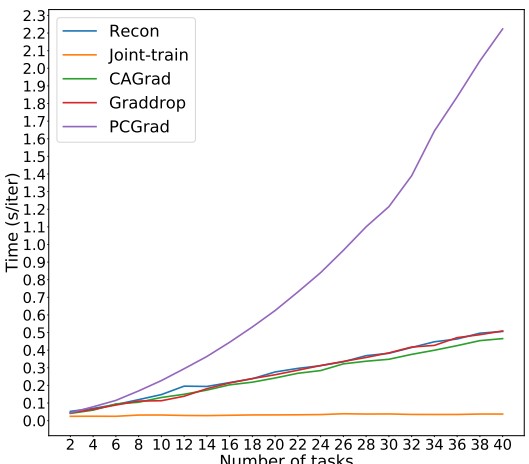

Figure 11: Comparison of running time (one iteration, excludes data fetching) on CelebA dataset.

**Analysis of running time.** We evaluate how Recon scales with the number of tasks on CelebA dataset, by comparing the running time of one iteration used by Recon in computing gradient conflict scores (the most time-consuming part of Recon) to that of the baselines. The results in Fig. 11 show that Recon is as fast as other gradient manipulation methods such as CAGrad (Liu et al., 2021a) and Graddrop (Chen et al., 2020), but much slower than joint-training especially when the number of tasks is large, which is natural since Recon needs to compute pariwise cosine similarity of task gradients. However, since Recon only needs to search for the conflict layers once for a given network architecture, as discussed above, the running time is not a problem.

