# OpenReview forum: "Recon: Reducing Conflicting Gradients From the Root For Multi-Task Learning"
_ICLR.cc/2023/Conference — ICLR 2023 poster_

### Official Review · Reviewer_Tqjk · 2022-10-24

**Confidence:** 5
**Correctness:** 4
**Technical Novelty And Significance:** 3
**Empirical Novelty And Significance:** 4
**Recommendation:** 8

**Clarity, Quality, Novelty And Reproducibility:**

The paper is well-written and proposes a novel approach. Although the approach is simple, reproducibility would be easier if code can be shared.

Definition 1 has a typo: “angel” -> angle


**Strength And Weaknesses:**

While the approach is straightforward, the simplicity and effectiveness of the proposed Recon method are appealing.

A potential concern with resolving conflicting gradients at training time is that they may, in some cases, provide a regularisation benefit that leads to positive transfer on held-out data. I would have liked to see more discussion on this issue.

The experiments are limited in two respects: (1) they focus only on vision problems; and (2) three out of four multi-task settings involve semantic segmentation. I would have liked to see more diversity of benchmarks as well as benchmarks with larger numbers of tasks.

A downside of the proposed approach is that the resulting model is not actually multi-task anymore, as it contains task-specific weights, which could be problematic in some settings for deployment.

Finally, the hyperparameters (S, K) appear to be quite sensitive, which increases the cost of the approach as they require fine-tuning for the different domains / datasets.


**Summary Of The Paper:**

This paper tackles conflicting gradients in multi-task learning (MTL). Specifically, the idea is to identify the shared layers of the multi-task model that exhibit large degrees of conflicting gradients, and replace these layers with task-specific parameters. The reported experiments suggest that only a small number of additional task parameters are necessary to improve performance, and reduce the occurrence of conflicting gradients in the remaining shared layers.

The experiments cover four multi-task benchmarks: multi-fashion+MNIST, CityScapes, NYUv2, and PASCAL-Context. The main finding is that the proposed “Recon” MLT approach improves the performance of all base models. Notably, the performance increase can be obtained with only a small increase in the total number of parameters (1.42%, 12.77%, 0.26%, and 9.8%, respectively for each benchmark).

Further analysis is provided in the way of an ablation study, showing that Recon significantly reduces the occurrence of conflicting gradients. Additionally, a theoretical analysis is presented, showing that a single gradient update on the model parameters of Recon achieves lower loss than on the original model parameters.


**Summary Of The Review:**

Overall, this paper describes a simple idea to improve MTL by introducing a relatively small number of task-specific parameters, in order to reduce the impact of gradient conflict. The experiments are fairly comprehensive.

---

> ### Author Response · Authors · 2022-11-19
> **Response to Reviewer Tqjk**
>
> Thank you for the positive feedback and helpful comments.
>
> **Q1: A potential concern with resolving conflicting gradients at training time is that they may, in some cases, provide a regularisation benefit that leads to positive transfer on held-out data. I would have liked to see more discussion on this issue.**
>
> **A1:** Thank you for the insightful comment. We agree that resolving conflicting gradients at training time may have a similar effect as regularisation. However, unlike previous gradient manipulation methods, our method Recon is designed to resolve conflicting gradients **before training, not during training.** In other words, it aims to resolve conflicts caused by multi-task architecture design. Notice that after finding the conflict layers and setting them task-specific, we **train the modified network from scratch** (we apologize for not making this clear in the previous version). The network modified with the found layers can be directly used by different MTL methods (e.g, joint-training, gradient manipulation methods, or architecture search methods) on different datasets to greatly reduce conflicting gradients (Tables 6 and 8-10) and significantly improve performance (Tables 1-5, 15, and 16).
>
> **Q2: I would have liked to see more diversity of benchmarks as well as benchmarks with larger numbers of tasks.**
>
> **A2:** Thanks for the advice.
> We've validated our method on an additional benchmark CelebA [1] with **40 classification tasks**. We follow [2] to formulate the problem as a 40-tasks learning problem, where each task is a binary classification task. The results are provided in **Table 2** and copied below, which show that even when dealing with a large number of tasks, our method Recon can still consistently improve various methods with a reasonable increase in model parameters.
>
> |        Method        | Single-task | Joint-train |  w/ Recon | CAGrad |  w/ Recon | Graddrop |  w/ Recon | PCGrad |  w/ Recon |
> |:--------------------:|:-----------:|:-----------:|:---------:|:------:|:---------:|:--------:|:---------:|:------:|:---------:|
> |     Average Error    |     8.38    |    8.33     | **8.22** |  8.31  | **8.23** |   8.33   | **8.20** |  8.64  | **8.36** |
> | $\Delta m\%\uparrow$ |      -      |    0.55     | **1.92** |  0.79  | **1.74** |   0.23   | **2.13** | -3.14  | **0.24** |
> |         \#P. (MB)         |   1706.03   |    43.26    |   68.03   |  43.26 |   68.03   |   43.26  |   68.03   |  43.26 |   68.03   |
>
> **Q3: A downside of the proposed approach is that the resulting model is not actually multi-task anymore, as it contains task-specific weights, which could be problematic in some settings for the deployment.**
>
> **A3:** We agree this is a limitation of our method. It'd be interesting to study whether it is possible to avoid conflicting gradients without using task-specific weights.
>
> **Q4: The hyperparameters (S, K) appear to be quite sensitive, which increases the cost of the approach as they require fine-tuning for the different domains / datasets.**
>
> **A4:** The hyperparameters (S, K) are used to find the conflict layers. While the performance of a network (e.g., ResNet18) is sensitive to S and K, the good thing is, for the same network, we may only need to tune them once since the found layers are transferable between different datasets, which helps to save the cost of re-tuning the hyperparameters. Please see our response to Q2 and Q4 of Reviewer DXpE and **Pages 19-20 in Appendix C.**
>
> **Q5: Definition 1 has a typo: “angel” -> angle.**
>
> **A5:** Thanks for pointing this out. We have corrected the typo.
>
> [1] Liu et al., Deep learning face attributes in the wild. CVPR 2015.
>
> [2] Sener et al., Multi-task learning as multi-objective optimization. NeurIPS 2018.

---

### Official Review · Reviewer_DXpE · 2022-10-24

**Confidence:** 5
**Clarity, Quality, Novelty And Reproducibility:** The paper is very clear and should be…
**Correctness:** 4
**Technical Novelty And Significance:** 3
**Empirical Novelty And Significance:** 3
**Recommendation:** 6

**Strength And Weaknesses:**

Strength.
+ The paper is very well written and clearly motivated, supported with a wide range of experiments.
+ The proposed method is general in design and can be attached to any optimisation methods and any architecture.
+  The proposed method can achieve smaller loss supported with theoretical analysis.

Limitations.
+ Scalability and Running Time. Clearly, one inevitable problem for the proposed method is to compute cosine similarity for all combination of task pairs which might be unscalable when the number of training tasks is large. Though I agree that the newly initialised task-specific parameters might be small compared to the total network parameters, but the running time might be much slower. It would be clearer if the authors could the list the increased running time for such method and how it scales with increasing number of tasks. One paragraph that related to this calming that “requires extra 25% of training iterations” is clearly not sufficient and definitely not fast considering the fact that we only 2 tasks.

+ Conflict from Architecture Design. It’s very interesting to see that the authors found out that the conflict layers are consistent across multiple methods. This observation makes me wonder that the conflict itself was mainly due to the multi-task architecture design. As such, I am wondering whether these conflicts would reduce if we re-train the model from scratch after we set these conflict layers to be task-specific. If so, the running time would not be a problem since we only need to run this method once.  Just to be clear, the consistency is from different methods on the same dataset or different methods on different datasets as well?
+ Severity criteria. Is this hyper-parameter needs to be tuned for each dataset as the paper lists the percentage of increased parameters vary quite a lot from different datasets, which will make the method a bit less general.
+ Minor visualisation suggestion. Please add the attached method before (w/ Recon) or have a midrule to separate other methods. This would make the comparison much clearer.


**Summary Of The Paper:**

This paper proposes a multi-task learning method to reduce conflict gradients during multi-task optimisation. The proposed method, name Recon is very simple in design and intuitive – it first computes the cosine similarity score for all combinations of task pairs in each shared layer, and sets this shared layer to be task-specific if its cosine similarity score is negative for multiple task pairs. As such, the proposed method can be combined with any gradient-based multi-task optimisation methods and can be applied on any multi-task architecture. The paper verifies this design on a range of multi-task benchmarks with different architectures and different optimisation methods.

**Summary Of The Review:**

The paper proposes a simple multi-task optimisation method: to make task-shared layers to be task-specific if the gradient conflict is severe. The paper in general is very well written with comprehensive empirical and theoretical support.

---

> ### Author Response · Authors · 2022-11-19
> **Response to Reviewer DXpE**
>
> Thank you for the positive feedback and constructive comments.
>
> **Q1:** **It’s very interesting to see that the conflict layers are consistent across multiple methods. It makes me wonder that the conflict itself was mainly due to the multi-task architecture design. As such, I am wondering whether these conflicts would reduce if we re-train the model from scratch after we set these conflict layers to be task-specific. If so, the running time would not be a problem since we only need to run this method once.**
>
> **A1:** Thanks for the insightful comment. **We did re-train the network from scratch** after turning the conflict layers task-specific (we apologize for not making this clear in the previous version). In this way, it can reduce conflicting gradients from the start of training. You completely get it right that the conflict is mainly caused by multi-task architecture design. As can be seen from Tables 6 and 8-10, after turning the conflict layers task-specific, the conflict gradients are greatly reduced (by more than 50\%) for different methods on different datasets.
>
> **Q2: Is the consistency from different methods on the same dataset or different methods on different datasets as well?**
>
> **A2:** Thanks for the thoughtful question. In our previous experiments, we found the consistency holds for different methods on the same dataset. Hence, we used joint-training to find the conflict layers and directly applied the modified network to improve various methods *without searching for the layers again*. Inspired by your question, we've conducted additional experiments and found the consistency also holds for different datasets.
>
> Specifically, *with the same network architecture, the conflict layers found on one dataset can be used to modify the network to be directly used on another dataset to consistently improve the performance of various baselines.* To verify this, we've conducted experiments with three different architectures: ResNet18, SegNet, and MTAN.
>
> For ResNet18, we find that the layers found on CelebA and those found on Multi-Fashion+MNIST are **exactly the same.** For SegNet, we find that **95\% layers (38 out of 40)** found on NYUv2 **are identical** to those found on CityScapes. For MTAN (SegNet with attention), we find that **68\% layers (17 out of 25)** found on CityScapes **are identical** to those found on NYUv2.
>
> On NYUv2, we compare the performance of using conflict layers found on CityScapes (by SegNet) with those found on NYUv2 (**Table 15**). On CityScapes, we compare the performance of using conflict layers found on NYUv2 (by MTAN) with those found on CityScapes (**Table 16**). The above results show that for the same network, the found conflict layers are transferable between different datasets, while searching for the layers again on the new dataset may lead to better performance for some networks.
>
> Please see more detailed discussion in **Pages 19-20 in Appendix C**.
>
> **Q3: The proposed method needs to compute cosine similarity for all combinations of task pairs which might be unscalable when the number of training tasks is large. It would be clearer if the authors could list the increased running time for such a method and how it scales with the increasing number of tasks.**
>
> **A3:** Thanks for the thoughtful comment. We evaluate how Recon scales with the number of tasks on CelebA which contains 40 tasks. We compare the running time of one iteration used by Recon in computing gradient conflict scores (the most time-consuming part) to that of the baselines. The results in **Fig. 11** (in Appendix C) show that Recon is as fast as other gradient manipulation methods such as CAGrad and Graddrop, but much slower than joint-training especially when the number of tasks is large, which is natural since Recon needs to compute pairwise cosine similarity of task gradients. However, as you mentioned, since it only needs to search for the conflict layers once, the running time is not a problem.
>
> **Q4: Does the severity criteria $S$ need to be tuned for each dataset? As the paper lists the percentage of increased parameters varies quite a lot from different datasets, which will make the method a bit less general.**
>
> **A4:** Please note that we use 4 different networks for different datasets of classification and segmentation tasks (i.e., ResNest18 for Multi-Fashion+MNIST, SegNet for CityScapes, MTAN for NYUv2, and MobileNetv2 for PASCAL-Context), which is the main reason the percentage of increased parameters varies a lot. For the same network (e.g., ResNet18), we may only need to tune the hyperparameters ($S$ and $K$) once to find the conflict layers. For example, the layers found on Multi-Fashion+MNIST with ResNet18 are completely transferable to the new dataset CelebA, so we can directly modify ResNet18 to use it for multi-task learning on CelebA.
>
> **Q5: Use midrule to separate different methods.**
>
> **A5:** Thanks for the nice suggestion. We've used midrule to separate different methods.

---

> > ### Comment · Reviewer_DXpE · 2022-11-26
> > **Response to the rebuttal**
> >
> > Thanks to the authors for the detailed rebuttal. The improved paper is very clear and the additional experiments are convincing. I would love to keep my weak acceptance as my final rating.

---

> > > ### Author Response · Authors · 2022-11-26
> > > **Thank you**
> > >
> > > We would like to thank you for your careful reading and constructive comments of our work! Your review has expanded our thinking and strengthened the manuscript.

---

### Official Review · Reviewer_2d79 · 2022-10-25

**Confidence:** 4
**Correctness:** 3
**Technical Novelty And Significance:** 3
**Empirical Novelty And Significance:** 3
**Recommendation:** 8

**Clarity, Quality, Novelty And Reproducibility:**

Clarity: this paper is clearly written with good organization.

Quality, Novelty: this paper probed an important issue of the existing methods and proposed original solutions.

Reproducibility: though the proposed method is fairly concise and should be easy to implement, the reproducibility still depends on if the authors release the codes.

**Strength And Weaknesses:**

Strength:

1. Important problem with the pilot study probing existing issues and the concise solution addressing them.
2. The theoretical analysis is also carried out that guarantees a lower training loss.
3. This paper is well-organized and easy to follow.

Weaknesses:

1. In order to avoid , there is another category of MTL methods aims to avoid the negative transfer/gradient conflict by designing proper network architectures. For example, [1-4] perform layerwise feature fusing between tasks, which do not consider splitting from a shared network, but starting with several individual networks to merge (fuse features). Can the authors discuss, and preferably compare with, those methods?
2. Would it help to control the starting iteration step when calculating (i.e., summing up) the s-scores? In other words, currently s^(k) = \sum_{i=1}^{I} s_i^(k), would it help if i does not start with 1 as the s-scores of the very initial iterations can be noisy?
3. This could be a minor issue: there exists a much larger MTL dataset, i.e., Taskonomy [5], to validate the proposed method.

[1] Misra et al., Cross-stitch networks for multi-task learning. CVPR 2016.

[2] Gao et al., NDDR-CNN: Layerwise feature fusing in multi-task CNNs by neural discriminative dimensionality reduction. CVPR 2019.

[3] Ruder et al., Latent multi-task architecture learning. AAAI 2019.

[4] Gao et al., MTL-NAS: Task-Agnostic Neural Architecture Search towards General-Purpose Multi-Task Learning. CVPR 2020.

[5] Zamir et al., Taskonomy: Disentangling task transfer learning. CVPR 2019.

**Summary Of The Paper:**

This paper proposes to mitigate the negative transfer, i.e., the conflicting gradients, of different tasks on the shared layers. After showing that the gradient surgery methods cannot reduce the occurrence of conflict by investigating the gradient angle distributions of different tasks, the authors propose to split those layers with the most conflicting gradients.

Specifically, the authors train the network for some initial steps to calculate an S score to indicate the gradient conflict, where for a specific layer, its S score is the number of tasks whose gradient angle is larger than a threshold. Then, the shared layer with a higher S score (i.e., higher gradients conflict) is split for T times for each of T tasks. Finally, the new network with split task-specific layers is continued to train until convergence. Theoretical analysis is also provided to guarantee that applying the proposed Recon reduces the training loss.

The experiments are conducted on Multi-Fashion+MNIST, CityScapes, NYUv2, and PASCAL-Context with ablations validating the promising performance.

**Summary Of The Review:**

Overall I think this is a good paper with clear (empirical) problem analysis and concise solution. For those further possible improvements, please see the Weakness section.

---

> ### Author Response · Authors · 2022-11-19
> **Response to Reviewer 2d79**
>
> Thank you for the positive feedback and thoughtful comments.
>
> **Q1: Compare Recon with MTL methods that design network architectures**
>
> **A1:** Compared with soft parameter sharing methods [1-4], Recon has much better scalability when dealing with a large number of tasks. For example, on the CelebA dataset with 40 tasks, Recon can consistently improve the performance of baselines with a reasonable increase in the total number of parameters (**57.25\%** as shown in **Table 2**), while soft parameter sharing methods such as Cross-Stitch will increase the model size by nearly **40 times**.
>
> We have included the discussion in the related work section. Moreover, we have included Cross-Stitch [1] (implemented using SegNet based on the code of CAGrad [5]) as a baseline for comparison on CityScapes (**Table 3**) and NYUv2 (**Table 5**) datasets.
>
>
> **Q2: Would it help to control the starting iteration step when calculating (i.e., summing up) the s-scores? In other words, currently, $s^{(k)} = \sum\_{i=1}^{I} s\_i^{(k)}$, would it help if I do not start with 1 as the s-scores of the very initial iterations can be noisy?**
>
> **A2:**
> Thanks for the insightful question. Since the network layers are ranked by S-conflict scores, we measure the differences in rankings obtained in different training stages (i.e., in the 1st, 2nd, 3rd, or 4th 25\% iterations) by **Eq. 24** on Multi-Fashion+MNIST. The results in **Table 12** indicate that the layers found in different training stages are very similar.
>
> Further, in **Table 13**, we compare the performance of the networks modified by Recon with conflict layers found in different training stages on CityScapes.
> The results show that the layers found in the later stages (i.e., 3rd or 4th 25\% iterations) lead to slightly better performance than those found in the early stages (i.e., 1st or 2nd 25\% iterations), indicating the conflict scores in early iterations might be a bit noisy, just as you suspected. However, since the performance gaps are acceptably small, to save time, we use the initial 25\% training iterations to find conflict layers.
>
> Please see more detailed discussion in **Page 18 in Appendix C**.
>
> **Q3: This could be a minor issue: there exists a much larger MTL dataset, i.e., Taskonomy, to validate the proposed method.**
>
> **A3:** Thanks for the suggestion. Due to limited time and resources, we are unable to provide results on Taskonomy [6]. Instead, we conduct experiments on another dataset CelebA [7]. Following [8], we formulate the problem as a 40-tasks learning problem, where each task is a binary classification task. The results are provided in **Table 2** and copied below, which show that even when dealing with a large number of tasks, our method Recon can still consistently improve
> various methods with a reasonable increase in model parameters.
>
> |        Method        | Single-task | Joint-train |  w/ Recon | CAGrad |  w/ Recon | Graddrop |  w/ Recon | PCGrad |  w/ Recon |
> |:--------------------:|:-----------:|:-----------:|:---------:|:------:|:---------:|:--------:|:---------:|:------:|:---------:|
> |     Average Error    |     8.38    |    8.33     | **8.22** |  8.31  | **8.23** |   8.33   | **8.20** |  8.64  | **8.36** |
> | $\Delta m\%\uparrow$ |      -      |    0.55     | **1.92** |  0.79  | **1.74** |   0.23   | **2.13** | -3.14  | **0.24** |
> |         \#P. (MB)         |   1706.03   |    43.26    |   68.03   |  43.26 |   68.03   |   43.26  |   68.03   |  43.26 |   68.03   |
>
>
>
> [1] Misra et al., Cross-stitch networks for multi-task learning. CVPR 2016.
>
> [2] Gao et al., NDDR-CNN: Layerwise feature fusing in multi-task CNNs by neural discriminative dimensionality reduction. CVPR 2019.
>
> [3] Ruder et al., Latent multi-task architecture learning. AAAI 2019.
>
> [4] Gao et al., MTL-NAS: Task-Agnostic Neural Architecture Search towards General-Purpose Multi-Task Learning. CVPR 2020.
>
> [5] Liu et al., Conflict-averse gradient descent for multi-task learning. NeurIPS 2021.
>
> [6] Zamir et al., Taskonomy: Disentangling task transfer learning. CVPR 2019.
>
> [7] Liu et al., Deep learning face attributes in the wild. CVPR 2015.
>
> [8] Sener et al., Multi-task learning as multi-objective optimization. NeurIPS 2018.

---

### Official Review · Reviewer_5G4Q · 2022-10-28

**Confidence:** 5
**Correctness:** 3
**Technical Novelty And Significance:** 3
**Empirical Novelty And Significance:** 3
**Recommendation:** 3

**Clarity, Quality, Novelty And Reproducibility:**

Clarity: Generally, this paper is easy to follow. But the key part of the algorithmic design is not clear. Readers will find it hard to understand what are the precise actions of "turn the layers into task-specific ones".

Quality and Novelty: Fairly good

Reproducibility: The clarity affects the reproducibility to some extent.

**Strength And Weaknesses:**

Strengths:

+ The perspective on identifying conflicting gradients at layer scale is interesting. The proposed method that turn layers with high conflicts to task-specific ones seems to work well.
+ The overall structure of this paper is easy to follow.

Weaknesses:
+ I am not convinced by the claim that "gradient manipulation can not reduce the occurrence of conflicting gradients". In my practice, I find these techniques can alleviate the conflicting gradients. Your results are only based on one dataset. It's surely not enough. Moreover, how do you calculate the conflicting angles? I think Figure 1 and Section 3.3 need more in-depth explanations. Otherwise, this claim seems to be exaggerated.
+ The key algorithmic design "turn the layers into task-specific ones" is hard to figure what its exact meaning. I struggled to understand this point very concretely. Some figures with network structures may help better illustrate this clearly.
+ The applicability of this method to other commonly used MTL structure such as MMoE is not provided.

**Summary Of The Paper:**

This paper provides a new perspective on solving the conflicting gradients problem in MTL. It checks the conflicting level at layer scale, and turns the layers with high conflict scores into task-specific ones. Some theoretical analysis and experimental verifications are provided.

**Summary Of The Review:**

The paper is in general interesting. But in the current form, some parts of this paper are questionable or not clear. The authors need to clarify two main points raised in the review.

(1) The claim on the effects  of gradient manipulation to conflicting gradients.

(2) What does "set these layers task-specific" mean?

(3) More experiments or discussions are needed to see whether the proposed applies to typical MTL structures such as MMoE.

If the concerns are well addressed, I will raise my score.

---

> ### Author Response · Authors · 2022-11-19
> **Response to Reviewer 5G4Q**
>
> Thank you for the thoughtful comments. We've followed your advice to provide more supporting evidence and revise the manuscript.
>
> **Q1:The support for the claim "gradient manipulation can not effectively reduce the occurrence of conflicting gradients" is insufficient. Moreover, how do you calculate the conflicting angles?**
>
> **A1**:
> 1) For each training iteration, we first calculate the task gradients of all tasks w.r.t. the shared parameters (i.e., $\mathbf{g}\_i$ for any task $i$) and compute the conflict angle between any two task gradients $\mathbf{g}\_i$ and $\mathbf{g}\_j$ in terms of $cos{\phi\_{ij}}$. We then count and draw the distribution of $cos{\phi\_{ij}}$ in all training iterations.
>
> 2) In the previous version, we have provided the statistics of gradient conflicts on two datasets: Multi-Fashion+MNIST (Figures 1, 5, and Table 9 in Appendix C) and CityScapes (Figure 3 and Table 8 in Appendix C). We apologize for the wrong caption of Table 8, it should be CityScapes instead of Multi-Fashion+MNIST.
>
> 3) In the current version, we've included statistics on two more datasets: NYUv2 and PASCAL-Context. We also provide detailed comparison of Recon with existing gradient manipulation methods on each dataset: Multi-Fashion+MNIST (**Figures 1, 4, and Table 6**), CityScapes (**Figures 5, 8, and Table 8**), NYUv2 (**Figures 6, 9, and Table 9**), and PASCAL-Context (**Figures 7, 10, and Table 10**).
>
> The results show that gradient manipulation methods cannot effectively reduce the occurrence of conflicting gradients (compared to joint-training). They slightly reduce it in some cases, while in some other cases they even increase it. In contrast, Recon can greatly reduce the occurrence of conflicting gradients (more than 50\%) in every case (i.e., for different methods on different datasets).
>
>
> **Q2: The key algorithmic design "turn layers into task-specific ones" is hard to understand.**
>
> **A2**: Thank you for letting us know about this issue. To illustrate the design, we've provided a graphic illustration in **Fig. 2 (d)** of the current version.
>
> **Q3: Does Recon applicable to some commonly used MTL structures such as MMoE?**
>
> **A3**:
> Yes, Recon can be applied to MMoE. MMoE includes a set of expert networks that are shared across tasks, and for each task, it has a gate module to fuse the output of experts. We treat the expert networks as shared and the gates as task-specific. We implement MMoE on Multi-Fashion+MNIST dataset. We use two ResNet18 networks as experts (see more implementation detail in **Appendix B.1**). The experiments are repeated over 3 random seeds, and the mean values are reported. We've included the results in **Table 1** and copied them below, which show Recon can further improve MMoE. Note that in Table 1, better results are achieved by Recon combined with PCGrad [1] and CAGrad [2] with much less parameters.
>
> |  Method  | T1 Acc |   T2 Acc  | $\Delta m$\% |  \#P. (MB)  |
> |:--------:|:------:|:---------:|:------------:|:------:|
> |   MMoE   |  98.27 |   89.51   |     -0.12    |  85.62 |
> | w/ Recon |  98.25 | **89.67** |   **-0.04**  | 105.70 |
>
> [1] Yu et al., Gradient surgery for multi-task learning. NeurIPS 2020
>
> [2] Liu et al., Conflict-averse gradient descent for multi-task learning. NeurIPS 2021.

---

> > ### Author Response · Authors · 2022-11-30
> > **Please let us know your thoughts**
> >
> > Dear Reviewer 5G4Q,
> >
> > We believe we’ve addressed all your concerns including clarity and more experiments. Please see our response and the revised manuscript where revisions are highlighted in blue.
> >
> > We’d be grateful if you could share your further feedback. Thank you for your valuable time and consideration.
> >
> > Regards,  Authors

---

> > ### Author Response · Authors · 2022-12-10
> > **Please let us know if you have further questions**
> >
> > Dear Reviewer  5G4Q,
> >
> > As the discussion period is closing soon, we would really appreciate if you would let us know whether your concerns have been resolved. We would be happy to discuss with you if you have further questions. Thank you very much for your time!
> >
> > Regards, Authors

---

### Author Response · Authors · 2022-11-19
**To All Reviewers**

We sincerely thank all the reviewers for their thoughtful and helpful comments, which help us to improve the manuscript substantially. In the current version, we've highlighted the revisions in blue. The main modifications include the following.

1. We have made it clear that our method only needs to search for the conflict layers **once**, because *the found conflict layers are consistent for different methods on the same dataset and transferable between different datasets, given the same network architecture*. In our previous experiments, we used the joint-training baseline to find the conflict layers and directly applied the modified network to improve various methods (i.e., PCGrad, MGDA, Graddrop, CAGrad, BMTAS) on the same dataset, without searching for the layers again. Inspired by the comments of Reviewer DXpE and Reviewer Tqjk, we've conducted additional experiments and found the layers are transferable between different datasets, i.e., the conflict layers found on one dataset can be used to modify the network to be directly applied on another dataset to gain performance improvement, though searching for conflict layers again on the new dataset may sometimes lead to better performance (see **Tables 15, 16**, and **discussion in Appendix C**).

2. We've validated our method Recon on an additional dataset CelebA, which contains **40 tasks**. The results provided in **Table 2** show that *even when dealing with a large number of tasks, Recon still can consistently improve state-of-the-art methods with a reasonable increase (57.25\%) in model parameters*.

3. We've provided detailed statistics on four datasets (Multi-Fashion+MNIST, CityScapes, PASCAL-Context, and NYUv2) to demonstrate that previous gradient manipulation methods cannot effectively reduce conflicting gradients (see **Tables 6, 8-10, and Figures 1, 4, 5-10**).

Below we provide a point-to-point response to each reviewer to further clarify some concerns.

---

### Decision · Program_Chairs · 2023-01-20

**Decision:**

Accept: poster

**Justification For Why Not Higher Score:**

Paper is not quite of an award quality, but the scores do indicate broad consensus that the idea is interesting and well-executed.

**Justification For Why Not Lower Score:**

Reviewers were generally positive, and critical reviewer's comments appear to be adequately addressed. From my reading the paper is indeed of a good quality.

**Metareview: Summary, Strengths And Weaknesses:**

The paper proposes a method to improve the performance of multi-task learning methods, by reducing the conflict amongst gradients for different tasks. The key findings are that naive "gradient surgery" may not effectively solve this problem, while a simple per-layer conflict resolution is effective.

Three reviewers found the paper to be interesting, with a simple idea shown to be effective on a broad range of empirical settings. The careful study of why existing strategies may fail was also found to be illuminating, and the overall writing was seen to be of good quality. One reviewer raised some questions regarding the ineffectiveness of gradient manipulation, and the potential applicability to MMoE. The author response on these points is, in my reading, satisfactory. Thus, I believe the paper is suitable for publication, and would be of interest to the community.

**Note From Pc:**

if the above contains the word "oral" or "spotlight" please see: "oral" presentation means -> notable-top-5% and "spotlight" means -> notable-top-25%. As stated in our emails, we are disassociating presentation type from AC recommendations